# IGF2 mRNA binding protein-2 is a tumor promoter that drives cancer proliferation through its client mRNAs IGF2 and HMGA1

Ning Dai[1,2,3]*, Fei Ji[1,4], Jason Wright[5], Liliana Minichiello[6], Ruslan Sadreyev[1,7], Joseph Avruch[1,2,3]*

[1]Department of Molecular Biology, Massachusetts General Hospital, Boston, United States; [2]Diabetes unit, Medical Services, Massachusetts General Hospital, Boston, United States; [3]Department of Medicine, Harvard Medical School, Boston, United States; [4]Department of Genetics, Harvard Medical School, Boston, United States; [5]Program in Medical and Population Genetics, Broad Institute of Harvard and MIT, Cambridge, United States; [6]Department of Pharmacology, University of Oxford, Oxford, United Kingdom; [7]Department of Pathology, Harvard Medical School, Boston, United States

*For correspondence:
ning@molbio.mgh.harvard.edu
(ND);
avruch@molbio.mgh.harvard.edu
(JA)

**Competing interests:** The authors declare that no competing interests exist.

**Abstract** The gene encoding the Insulin-like Growth Factor 2 mRNA binding protein 2/IMP2 is amplified and overexpressed in many human cancers, accompanied by a poorer prognosis. Mice lacking IMP2 exhibit a longer lifespan and a reduced tumor burden at old age. Herein we show in a diverse array of human cancer cells that IMP2 overexpression stimulates and IMP2 elimination diminishes proliferation by 50–80%. In addition to its known ability to promote the abundance of Insulin-like Growth Factor 2/IGF2, we find that IMP2 strongly promotes IGF action, by binding and stabilizing the mRNA encoding the DNA binding protein *HMGA1*, a known oncogene. HMGA1 suppresses the abundance of IGF binding protein 2/IGFBP2 and Grb14, inhibitors of IGF action. IMP2 stabilization of *HMGA1* mRNA plus IMP2 stimulated IGF2 production synergistically drive cancer cell proliferation and account for IMP2's tumor promoting action. IMP2's ability to promote proliferation and IGF action requires IMP2 phosphorylation by mTOR.
DOI: https://doi.org/10.7554/eLife.27155.001

## Introduction

The insulin-like growth factor 2 (IGF2) mRNA-binding proteins (IGF2BP1-3 or IMP1-3) are a family of RNA binding proteins that participate in post-transcriptional gene regulation (*Nielsen et al., 1999*; *Yisraeli, 2005*); notably, SNPs in the second intron of the human IMP2 gene are associated with increased risk for Type 2 Diabetes (*Saxena et al., 2007*). IMPs enable translation of RNAs containing the IGF2 leader 3 5'UTR by internal ribosomal entry (*Dai et al., 2011*) and also bind to the IGF2 mRNA 3'UTR; IMP1 but not IMP2 participates in IGF2 RNA splicing (*Dai et al., 2013*). The IMPs contain six RNA binding motifs, two RRM domains followed by four KH domains and each are substrates for mTOR, which phosphorylates one (IMP1/3) (*Dai et al., 2013*) or two (IMP2) (*Dai et al., 2011*) serine residues in the segment that links the second RRM domain with the first KH domain. In RD rhabdomyosarcoma cells, the concurrent dual phosphorylation of IMP2 Ser162 and Ser164 is inhibited by rapamycin coincident with inhibition of IMP2 binding to the IGF2 leader 3 5'UTR (*Dai et al., 2011*) and IGF2 leader 3 mRNA translation by internal ribosomal entry.

**eLife digest** Some types of cancers develop when genes known as oncogenes or tumor promoters become faulty, and are present at abnormally high levels or inappropriately turned on. For example, cancer cells often have extra copies of the gene *IMP2* and therefore produce too much the IMP2 protein. Previous research has shown that mice that lack the IMP2 protein develop fewer cancers and live longer, while patients whose cancers make too much IMP2 have a poorer prognosis.

In healthy cells, the IMP2 protein normally helps to make new gene products by stabilising certain newly produced RNA molecules – the precursors of proteins, and in some cases by promoting the translation of these RNAs into proteins. For example, IMP2 binds to the mRNA that encodes the protein IGF2, which is a protein that helps cells to grow and is commonly produced in large quantities by cancer cells. However, until now it was not clear whether IMP2 only acts by increasing the production of IGF2 or also contributes to cancer growth in other ways.

Using a range of human cancer cell lines, and healthy mouse cells, Dai et al. first confirmed that without IMP2, cancer cells made less IGF2 and grew less quickly. When IGF2 was added to the cells lacking IMP2, it only partially restored their ability to grow. Further experiments revealed that cells without IMP2 had increased levels of proteins that counteract the effects of IGF2. Usually, IMP2 binds and stabilizes the mRNA that encodes the oncogenic protein HMGA1, which is known to regulate the number of 'anti-IGF2 proteins'. However, without IMP2, the HMGA1 levels drop, which causes an increase of the anti-IGF2 proteins.

This indicates that IMP2 promotes cancer cell growth both by enabling cells to produce more IGF2 and by suppressing inhibitors of IGF2 action. This suggests that cancer patients whose tumors have abnormally high levels of IMP2 may be especially sensitive to drugs that target and inhibit IGF2.

DOI: https://doi.org/10.7554/eLife.27155.002

In the mouse embryo, all three *IMP*s are expressed coordinately starting ~e10.5 coincident with the onset of IGF action; the expression of *Igf2, Imp1* and *Imp3* is largely extinguished before birth (*Nielsen et al., 1999*), whereas *Imp2* is widely expressed postnatally (*Dai et al., 2011*). Despite their architectural and sequence similarity, functional differences between the IMPs exist, as displayed most emphatically by the phenotypes of *Imp*-deficient mice. *Imp1* null mice are ~40% smaller than wildtype with aberrant intestinal development and ~50% mortality at p3 (*Hansen et al., 2004*). *Imp1* null mouse embryo fibroblasts (MEFs) exhibit deficient *Igf2* RNA splicing and translation and greatly slowed proliferation; the latter is rescued entirely by exogenous IGF2. In contrast, *Imp2* null mice are nearly normal in size through weaning, lean and slightly small as adults, highly resistant to diet-induced obesity and long lived (*Dai et al., 2015*). Investigating the prolonged lifespan of *Imp2* deficient mice, necropsy of an apparently healthy cohort at ~845–850 d age revealed the presence of malignant tumors in 4/6 $Imp2^{+/+}$ mice but in 0/6 $Imp2^{-/-}$ mice (*Dai et al., 2015*), raising the possibility that IMP2 contributes to tumorigenesis.

Herein we demonstrate that although the oncofetal IMPs are commonly reexpressed in human cancers, *IMP2* is usually much more abundant in most human cancers than its paralogs *IMP1* or *IMP3* (*Bell et al., 2013*, *Lederer et al., 2014*); moreover, the *IMP2* gene is amplified at a high frequency in several common solid tumors, a phenomenon rarely seen with the *IMP1* or *IMP3* genes. We show that IMP2 overexpression promotes, and IMP2 deficiency strongly inhibits the proliferation of both MEFs and an array of human tumor-derived cell lines. Beyond its known ability to promote *IGF2* translation, IMP2 controls the abundance of the oncogenic transcriptional regulator HMGA1 (*Fedele and Fusco, 2010*; *Ozturk et al., 2014*; *Sumter et al., 2016*) by binding and stabilizing *HMGA1* mRNA. In turn, HMGA1, another oncofetal protein, suppresses the transcription of *Igfbp2*, a high affinity extracellular IGF binding protein (*Diehl et al., 2009*; *Hoeflich et al., 1999*), and also reduces Grb14, an inhibitor of IGF1R and Insulin Receptor signaling (*Desbuquois et al., 2013*). IMP2's stabilization of *HMGA1* mRNA together with its stimulation of *IGF2* mRNA translation act synergistically to promote cell proliferation through mitogenic signaling by the IGF1R and the type A Insulin Receptor.

## Results

### IMP2 is widely overexpressed in human cancers

Data generated by the TCGA research network (http://cancergenome.nih.gov/) indicates that amplification of the *IMP2* gene is a relatively common event in comparison to amplification of *IMP1* and *IMP3* (*Figure 1A*), occurring in ~35–50% of squamous lung cancers, ~15–27% of ovarian cancers and in 15–20% of head and neck, esophageal, cervical and uterine cancers. Moreover, the absolute abundance of *IMP2* mRNA in all but a few cancers far exceeds that of the *IMP1* and *IMP3* paralogues (*Figure 1B*), even in those cancers wherein the fold amplification of *IMP1/IMP3* RNA over their level in the normal tissue is much greater than that of *IMP2*. Thus, *IMP2* is nearly always the most abundant *IMP* paralogue in human cancers and its overexpression occurs at a high frequency.

### IMP2 overexpression enhances and IMP2 deletion reduces proliferation of human cancer cell lines and mouse embryo fibroblasts (MEFs)

IMP2 polypeptide was overexpressed in several cancer-derived cell lines and in wildtype mouse embryo fibroblasts (MEFs). In each instance, IMP2 overexpression increased proliferation in a dose-dependent manner (*Figure 1C*). Reciprocally, we used the CRISPR-Cas9 to inactivate the *IMP2* gene in a diverse cohort of human cancer cell lines; parental controls used a GFP-directed guide. Each of the lines lacking IMP2 expression showed a substantial (52–78%) reduction in the rate of proliferation (*Figure 1D*). Similarly, *Imp2*$^{-/-}$ MEFs uniformly displayed less rapid growth, proliferating at ~19–25% the rate of *Imp2*$^{+/+}$ cells (*Figure 1E*). Seeking the mechanism(s) by which loss of IMP2 slows the proliferation we chose to focus on MEFs, which are less genetically heterogeneous than the cancer cells, but comparably responsive to IMP2 overexpression and elimination. IGF2 production by *Imp2*$^{-/-}$ MEFs is reduced by about 19% compared with *Imp2*$^{+/+}$ MEFs controls (*Figure 1F*), however saturating amounts of exogenous IGF2 do not rescue the proliferation of *Imp2*$^{-/-}$ MEFs (*Figure 1G*). This contrasts sharply with the response of *Imp1*$^{-/-}$ MEFs to IGF2, which restores their proliferative rate to 100% the level of *Imp1*$^{+/+}$ MEFs (*Dai et al., 2013*). Thus, the modest decrease of IGF2 production caused by *Imp2* inactivation contributes minimally to the slowed proliferation of *Imp2*$^{-/-}$ MEFs.

We applied deep sequencing to RNA extracted from lysates and from anti-IMP2 immunoprecipitates from *Imp2*$^{+/+}$ and *Imp2*$^{-/-}$ MEFs (*Supplementary file 1*). We also subjected triplicate aliquots of *Imp2*$^{-/-}$ and *Imp2*$^{+/+}$ MEFs to whole cell mass spectroscopic analysis using Tandem Mass Tag (TMT) Technology which yielded the quantitative estimation of the abundance of 7964 polypeptides (*Supplementary file 2*). The bioinformatic analysis of the RNAs enriched in the IMP2 immunoprecipitates (*Huang et al., 2009*), as well as the comparisons of the RNAome and proteome of *Imp2*$^{+/+}$ and *Imp2*$^{-/-}$ MEFs by GSEA (*Subramanian et al., 2005*) did not point to dominant mechanisms or specific elements that might underlie the slowed proliferation of the *Imp2*$^{-/-}$ MEFs. Seeking polypeptides involved in growth factor signaling and cell cycle progression whose altered abundance (*Supplementary file 2*) might underlie the slower proliferative rate of the *Imp2*$^{-/-}$ MEFs, we observed that the abundance of IGFBP2 (*Diehl et al., 2009*; *Hoeflich et al., 1999*) and Grb14 (*Desbuquois et al., 2013*) polypeptides, potential inhibitors of IGF1R and insulin receptor-A signaling, are increased 24.7- and 8.6-fold respectively in the *Imp2*$^{-/-}$ MEF proteome; Among cell cycle components, the largest difference is a 4.5 fold greater abundance of the cdk inhibitor Cdkn1a/p21Cip1 (*Warfel and El-Deiry, 2013*) in the *Imp2*$^{-/-}$ MEFs. The differential abundance of IGFBP2, Grb14 and p21Cip1 was confirmed by immunoblot, whereas the abundance of IMP1 and IMP3 proteins is not altered by deletion of *Imp2* (*Figure 2A*).

### Depletion of Grb14 and IGFBP2 each partially rescue *Imp2*$^{-/-}$ MEF proliferation but depletion of p21Cip1 does not

Lentiviral encoded, doxycycline-inducible shRNAs against *Grb14*, *Igfbp2* and *p21Cip1* were each stably expressed in both *Imp2*$^{+/+}$ and *Imp2*$^{-/-}$ MEFs. Induction of shRNA expression reduced the abundance of the target polypeptides progressively over the initial 4 days (*Figure 2B*), with levels in the *Imp2*$^{-/-}$ MEFs becoming almost as low as in the *Imp2*$^{+/+}$ MEFs by day 3 (*Figure 2C*). As regards proliferation, *Imp2*$^{+/+}$ MEFs containing shRNA against *Grb14* proliferated at the same rate whether treated with DMSO or doxycycline, increasing in number ~10 fold over seven days; *Imp2*$^{-/-}$ MEFs containing shRNA against *Grb14* treated with DMSO increased in number by ~3.5 fold, however

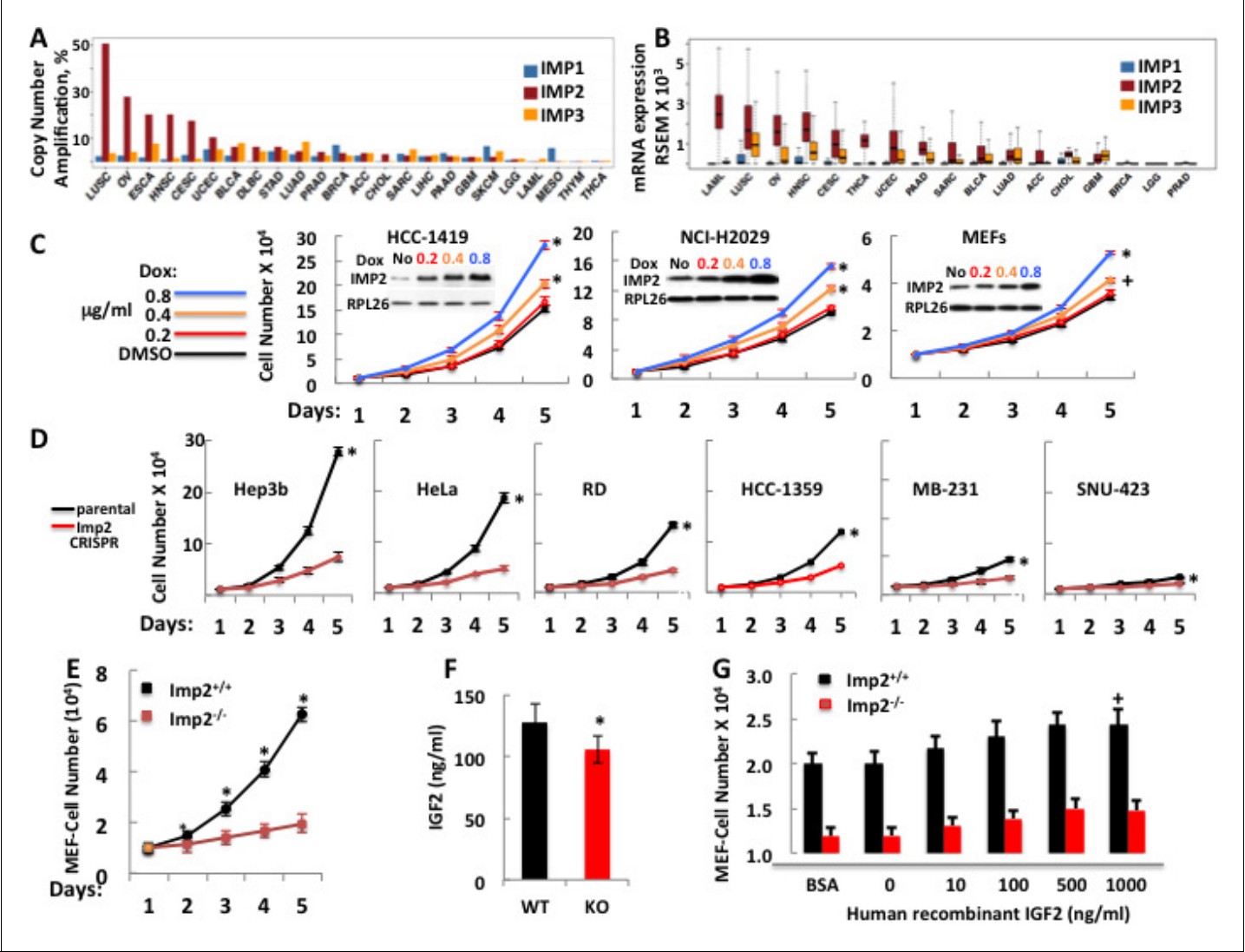

**Figure 1.** The *IMP2* gene is amplified and overexpressed in many cancers and drives proliferation. (A) The frequency of *IMP1,2,3* gene copy number amplification in various cancers. Data from TCGA. (B) *IMP1, 2* and *3* mRNA levels in various cancers. Data from TCGA. (C) IMP2 overexpression enhances proliferative rate. A vector encoding an IMP2 cDNA downstream of a doxycycline sensitive promoter was stably expressed in HCC-1419, NCI-H2029 and MEFs. Cells were treated with Doxycycline at the doses indicated and cell number was determined daily. +p<0.05, *p<0.01 vs DMSO. (D) CRISPR-mediated inactivation of the *IMP2* genes slows proliferation. The Hep3b, HeLa, RD, HCC-1359, MB-231 and SNU-423 cell lines were transfected with Cas9/CRISPR and a guide RNA directed at either *GFP* (black) or *IMP2* (red) sequences. Unselected polyclonal cell mixtures were plated in replicate and cell number was determined daily. *p<0.01 vs Imp2 CRISPR. (E) *Imp2$^{-/-}$* MEFs proliferate more slowly than *Imp2$^{+/+}$* MEFs. Littermate embryos from *Imp2$^{+/-}$* ± were harvested at e12.5–13.5 to derive *Imp2$^{+/+}$* and *Imp2$^{-/-}$* MEFs. Polyclonal mixtures were plated in replicate at passage 4 and cell number was determined daily. *p<0.01 vs Imp2$^{-/-}$. (F) *Imp2$^{-/-}$* MEFs produce less medium IGF2 than *Imp2$^{+/+}$* MEFs. Aliquots of the medium were taken at day 3 from the MEF cultures in *Figure 1E* and assayed for IGF2 polypeptide. *p<0.01 vs *Imp2$^{+/+}$*. (G) Supramaximal IGF2 does not restore the slower proliferation of *Imp2$^{-/-}$* MEFs to that of *Imp2$^{+/+}$* MEFs. *Imp2$^{+/+}$* and *Imp2$^{-/-}$* MEFs were plated in replicate in standard culture medium with the addition of BSA (1 ug/ml) or various amounts of human recombinant IGF2 and cell number was determined 48 hr later. +p<0.05 vs BSA.

DOI: https://doi.org/10.7554/eLife.27155.003

treatment with doxycycline increased *Imp2$^{-/-}$* MEF numbers ~8 fold over the same interval, that is, they proliferated at ~80% the rate of the *Imp2$^{+/+}$* MEFs (*Figure 2D*, bottom).

A similar response is observed with depletion of IGFBP2; doxycycline treatment of *Imp2$^{+/+}$* MEFs containing shRNA against *Igfbp2* increased proliferation slightly (~20%) over treatment with DMSO whereas doxycycline treatment of *Imp2$^{-/-}$* MEFs containing shRNA against *Igfbp2* increased cell number from ~30% to ~68% that of similarly-treated *Imp2$^{+/+}$* MEFs. (*Figure 2D*, second from

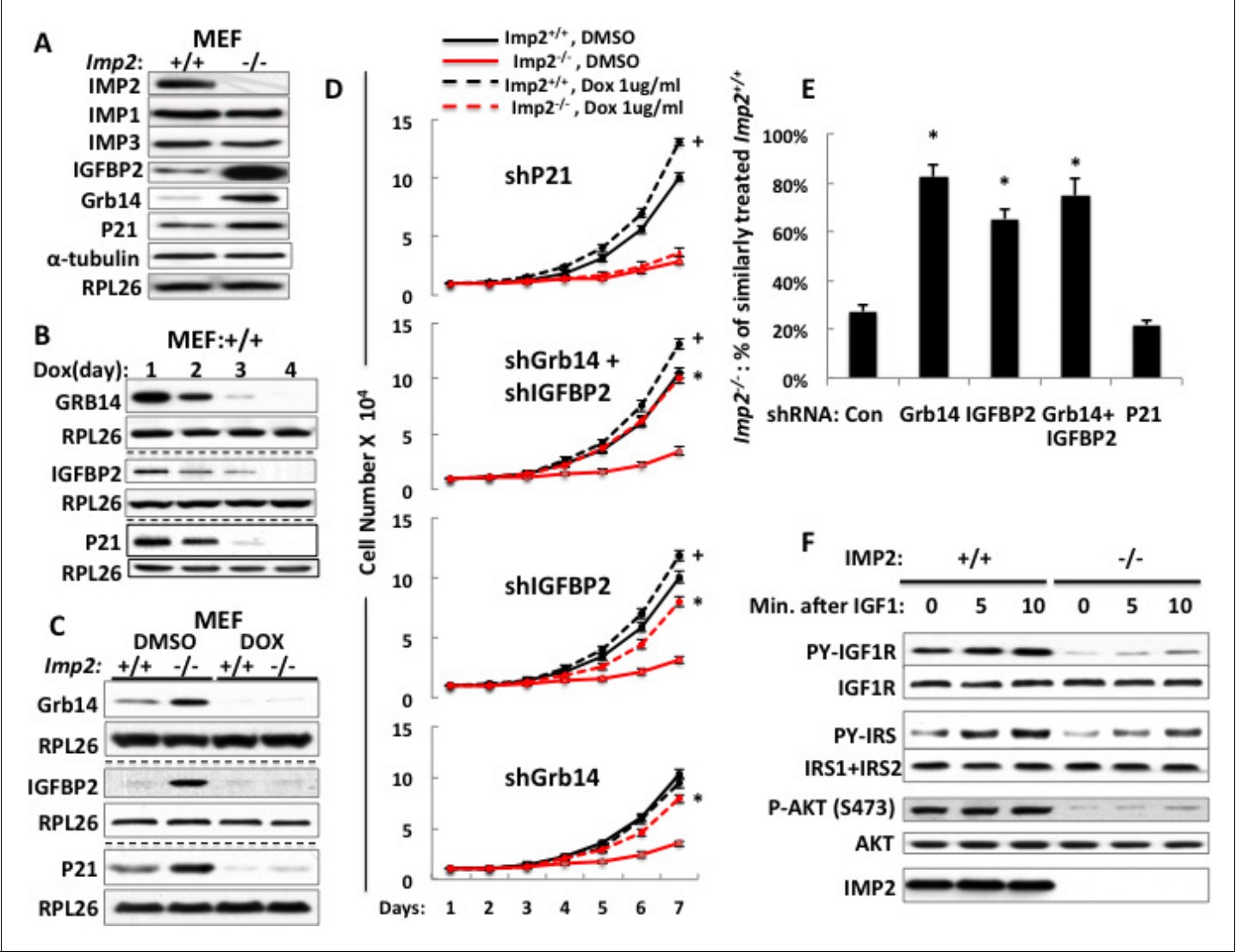

**Figure 2.** Overexpression of the Grb14 and IGFBP2 polypeptides is mostly responsible for the slowed proliferation of $Imp2^{-/-}$ MEFs. (**A**) $Imp2^{-/-}$ MEFs exhibit increased abundance of the IGFBP2, Grb14 and P21Cip1 polypeptides as compared with $Imp2^{+/+}$ MEFs but unaltered levels of IMP1 and IMP3. (**B**) Time course of shRNA-mediated depletion of *Grb14, Igfbp2* and *P21Cip1* from $Imp2^{+/+}$ MEF. (**C**) The effect of shRNA expression on the Grb14, IGFBP2 and P21Cip1 polypeptides in $Imp2^{+/+}$ and $Imp2^{-/-}$ MEFs. Polyclonal populations of $Imp2^{+/+}$ and $Imp2^{-/-}$ MEFs stably expressing shRNAs directed at *Grb14, Igfbp2* or *P21Cip1* were extracted 72 hr after addition of doxycycline. (**D**) Depletion of the Grb14 and IGFBP2 increases proliferation of $Imp2^{-/-}$ MEFs preferentially as compared with $Imp2^{+/+}$ MEFs. Polyclonal populations of $Imp2^{+/+}$ (black) and $Imp2^{-/-}$ (red) MEFs stably expressing shRNAs directed at *Grb14, Igfbp2, Grb14* and *Igfbp2* or *P21Cip1* were plated in replicate and DMSO (solid) or doxycycline (dashed) was added eight hours thereafter. The cell number was determined following day (=day 1) and daily thereafter. + = $p < 0.05$, Dox vs DMSO, $Imp2^{+/+}$; *=$p < 0.01$ Dox vs DMSO, $Imp2^{-/-}$. (**E**) The shRNAs against *Grb14, Igfbp2* or both restore the proliferation of $Imp2^{-/-}$ MEFs close to that of the correspondingly treated $Imp2^{+/+}$ MEFs, but *p21Cip1* shRNA does not. The increase in cell number at day 7 as shown in ***Figure 2D*** for $Imp2^{+/+}$ and $Imp2^{-/-}$ MEFs was recalculated as follows: the bar labeled 'Con' represents the increase in cell number at day 7 of $Imp2^{-/-}$ MEFs treated with DMSO divided by the increase in cell number at day 7 of DMSO-treated $Imp2^{+/+}$ MEFs, multiplied by 100. The other bars represent the same calculation for the doxycycline-treated $Imp2^{+/+}$ and $Imp2^{-/-}$ MEFs stably expressing shRNAs directed at *Grb14, Igfbp2, Grb14* and *Igfbp2* or *P21Cip1*. *=$p < 0.01$ vs Con. (**F**) IGF1R signaling is impaired in $Imp2^{-/-}$ MEFs. $Imp2^{+/+}$ and $Imp2^{-/-}$ MEFs, serum starved overnight were rinsed in serum-free DMEM and stimulated immediately thereafter by addition of human recombinant IGF1 (30 nM). Extracts were prepared before and at the times shown thereafter; the extracts and IPs of IGF1R and IRS1 +IRS2 therefrom were subjected to immunoblots as indicated.

DOI: https://doi.org/10.7554/eLife.27155.004

bottom, 2E). Concurrent depletion of Grb14 and IGFBP2 from $Imp2^{-/-}$ MEFs restored their proliferation to ~75% that of similarly treated $Imp2^{+/+}$ MEFs (*Figure 2D*, second from top, 2E). In contrast, depletion of p21Cip1 from $Imp2^{+/+}$ and $Imp2^{-/-}$ MEFs increased their proliferation by 1.33 and 1.36 fold respectively (*Figure 2D*, top, 2E). Thus, reversal of the 4.5 fold increased abundance of p21Cip1does not enable any rescue of $Imp2^{-/-}$ MEF proliferation.

We hypothesize that overexpression of Grb14 and IGFBP2 slow proliferation of $Imp2^{-/-}$ MEFs through inhibition of IGF1R and Type A insulin receptor-initiated signal transduction. To evaluate the signaling capacity of the IGF1R, $Imp2^{+/+}$ and $Imp2^{-/-}$ MEFs were deprived of serum overnight, rinsed and incubated in fresh serum-free medium and stimulated briefly with IGF1 (30 nM); although the abundance of the IGF1R and InsR beta subunits does not differ, the IGF1-induced increase in IGF1R beta subunit tyrosine phosphorylation in $Imp2^{-/-}$ MEFs is substantially reduced as compared with $Imp2^{+/+}$ MEFs, as is the overall tyrosine phosphorylation of IRS1 +IRS2 and activation of Akt assessed by phosphorylation at Ser473 (*Figure 2F*). Thus IGF1R signaling is impeded in the $Imp2$ null MEFs.

## Increased *Igfbp2* gene transcription and reduced Grb14 polypeptide degradation underlies their overabundance in $Imp2^{-/-}$ MEFs

Surprisingly, the *Igfbp2* and *Grb14* mRNAs are not among the 636 mRNAs 1.5-fold or more enriched in the IMP2 immunoprecipitates (*Supplementary file 1*) from $Imp2^{+/+}$ MEFs, i.e., *Igfbp2*- and *Grb14*-mRNAs are not IMP2 clients. The abundance of *Igfbp2* mRNA in $Imp2^{-/-}$ MEFs is increased 18.6 (RNAseq; *Supplementary file 1*) to 21.7 (by QPCR; *Figure 3A*, top) fold and its half life is prolonged ~1.7 fold (10.1 hr vs 5.9 hr) over $Imp2^{+/+}$ MEFs (*Figure 3B*, top); although there is also some increase in polysomal abundance (*Figure 3C*, top), the markedly increased abundance of the IGFBP2 polypeptide in $Imp2^{-/-}$ MEFs is attributable primarily to increased *Igfbp2* gene transcription. In contrast, *Grb14* mRNA is approximately 70% lower (RNAseq: *Supplementary file 1*; QPCR: *Figure 3A*,middle) and its half-life is ~40% shorter (3.89 hr vs 6.03 hr; *Figure 3B*, middle) in $Imp2^{-/-}$ versus $Imp2^{+/+}$ MEFs. Although there is a ~ 2.5 fold increase in *Grb14* mRNA polysomal abundance in the $Imp2^{-/-}$ MEFs (*Figure 3C*, middle), the primary mechanism for ~9 fold increased polypeptide abundance is post-translational, a marked prolongation of Grb14 polypeptide half-life from ~2 hr to ~31 hr (*Figure 3D*, middle); p21Cip1 follows a pattern generally similar to Grb14 (*Figure 3A–E*, bottom).

## Addition of IGF2 or depletion of Grb14 from IMP2 deficient human cancer cells partially restores their proliferation toward parental levels, often additively

As in MEFs, deletion of *IMP2* from the cancer cell lines is accompanied by a marked increase in the abundance of Grb14 and IGFBP2 (*Figure 4A*), and overexpression of IMP2 reduces the abundance of both the IGFBP2 and Grb14 polypeptides (*Figure 4B*). Notably, *IMP2* deletion from the human cancer cells causes a much greater decrease in IGF2 production than in MEFs, ranging from ~35–50% of parental levels (*Figure 4C*). Consequently, we attempted to overcome both the relative deficiency of IGF2 as well as the ability of IGFBP2 overabundance to sequester IGF2 by the addition of an excess of IGF2. The effect of adding exogenous IGF2 and shRNA-induced Grb14 depletion, singly and in combination, on the proliferation of the parental and IMP2 deficient cancer cell lines shown in *Figure 4D*. Depletion of Grb14 from parental cancer cells increased proliferation by only 1.04–1.12 fold whereas Grb14 depletion from the IMP2-deficient variants increased their proliferation by 1.5–2.09 fold. Similarly, addition of saturating amounts of IGF2 to the parental and IMP2 deficient cancer cells causes a substantially greater fractional increase in cell number in each of the IMP2 deficient variants than in the parental cells (except IMP2 deficient SNU-423, bottom right). Excess IGF2 alone however, like Grb14 depletion, restores total cell number of the IMP2 deficient variants only partially toward the parental levels; in the IMP2 deficient HeLa (*Figure 4D*, middle left), MB-231 (*Figure 4D*, middle right) and Hep3b cells (*Figure 4D*, top left), combining depletion of Grb14 with the addition of IGF2 restores cell numbers of the IMP2 deficient variants to 72%, 79% and 96% of the parental levels. Thus in these three lines the large majority of the IMP2-stimulated proliferation is mediated by enhancing the production and action of IGF2. The IMP2 deficient variants of RD (*Figure 4D*, bottom left) and HCC1359 cells (*Figure 4D*, top right) exhibit a more

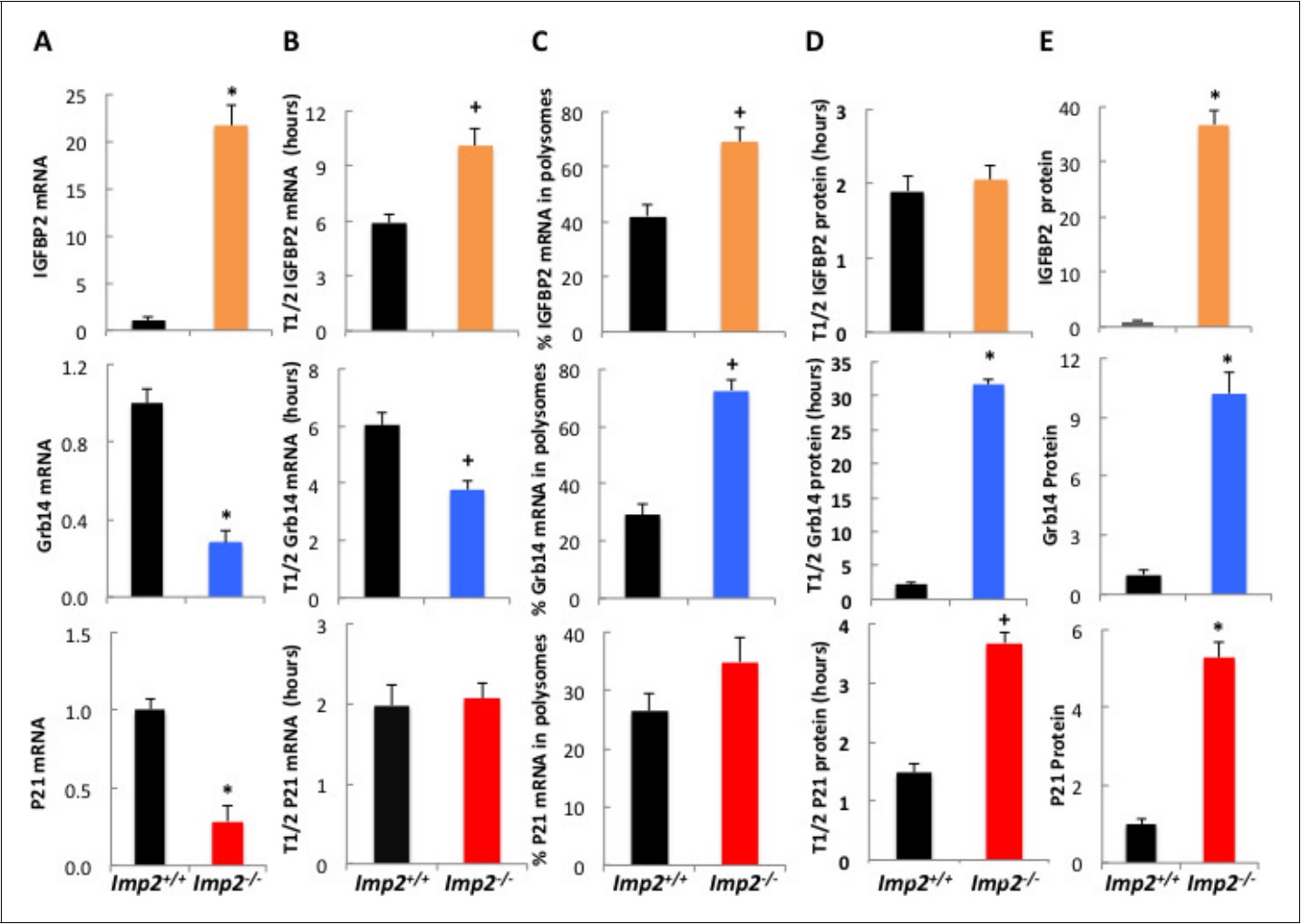

**Figure 3.** IGFBP2 polypeptide overabundance in *Imp2*$^{-/-}$ MEFs is due to increased gene transcription whereas Grb14 polypeptide overabundance is due to markedly slower polypeptide degradation. (**A**) *Igfbp2, Grb14* and *P21* mRNA abundance in *Imp2*$^{+/+}$ and *Imp2*$^{-/-}$ MEFs. RNA was extracted from *Imp2*$^{+/+}$ and *Imp2*$^{-/-}$ MEFs and the abundance of *Grb14, Igfbp2, P21, Gapdh* and *Actb* mRNA was determined by QPCR. The difference in Ct value between the *Imp2*$^{+/+}$ and *Imp2*$^{-/-}$ MEFs was converted to a numerical fraction; the value for the abundance of the *Imp2*$^{+/+}$ MEFs was set to 1.0 (black bars) and the relative abundance of the *Imp2*$^{-/-}$ MEFs is shown in colored bars. Values determined by genome-wide RNAseq are found in **Supplementary file 2**. (**B**) The turnover of *Igfbp2, Grb14* and *P21* mRNAs in *Imp2*$^{+/+}$ and *Imp2*$^{-/-}$ MEFs. Actinomycin 1 uM was added to rapidly growing *Imp2*$^{+/+}$ (black) and *Imp2*$^{-/-}$ (colored) MEFs; RNA was extracted at 0,2,4,6,8,10, 12 hr thereafter and the abundance of *Grb14, Igfbp2, P21, Gapdh* and *Actb* was determined by QPCR. The rate of decline was determined by least squares and the time representing a 50% decrease of the initial value is shown. (**C**) The percentage of *Igfbp2, Grb14* and *P21* mRNAs residing in polysomes in *Imp2*$^{+/+}$ and *Imp2*$^{-/-}$ MEFs. Total RNA and post-mitochondrial extracts were prepared from equal numbers of rapidly growing *Imp2*$^{+/+}$ (black) and *Imp2*$^{-/-}$ (colored) MEFs; the post-mitochondrial extracts were subjected to sucrose density gradient centrifugation. Total RNA and RNA extracted from the pooled polysomal fractions was quantified by QPCR and the ratio of polysomal/total RNA X 100 was determined for *Grb14, Igfbp2* and *P21*. (**D**) The turnover of the IGFBP2, Grb14 and P21 polypeptides in *Imp2*$^{+/+}$ and *Imp2*$^{-/-}$ MEFs. Cycloheximide 250 µM, was added to rapidly growing *Imp2*$^{+/+}$ (black) and *Imp2*$^{-/-}$ (colored) MEFs; extracts prepared at t = 0 and every hour for 12 hr were subjected to SDS-PAGE and immunoblot for Grb14, IGFBP2 and P21. Relative abundance was determined by densitometry; the rate of decline was calculated as in 3B. (**E**) The abundance of the IGFBP2, Grb14 and P21 polypeptides in *Imp2*$^{+/+}$ and *Imp2*$^{-/-}$ MEFs. Extracts prepared from rapidly growing *Imp2*$^{+/+}$ and *Imp2*$^{-/-}$ MEFs were subjected to SDS-PAGE and the relative abundance of Grb14, IGFBP2 and P21 was determined by immunoblot; the values for *Imp2*$^{+/+}$ were set to 1.0 (black) and the relative value for *Imp2*$^{-/-}$ is shown in the colored bars. Values determined by MS are found in **Supplementary file 2**. For all sections, $+p<0.05$, $*=p < 0.01$ vs *Imp2*$^{+/+}$.

DOI: https://doi.org/10.7554/eLife.27155.005

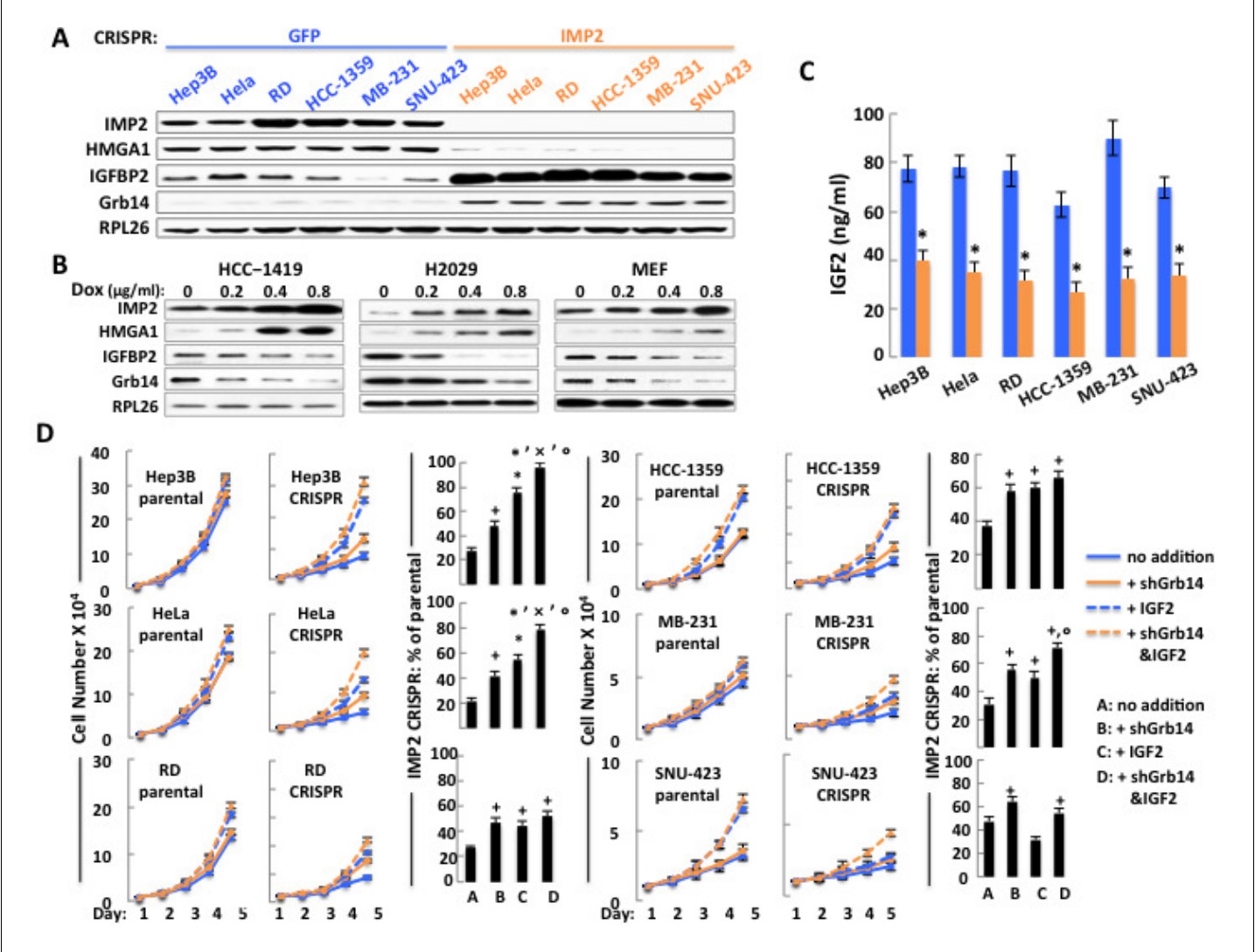

**Figure 4.** The ability of IMP2 to drive the proliferation of human cancer cells is mostly due to IMP2 regulation of IGF2, Grb14 and IGFBP2 abundance. (A) CRISPR mediated deletion of *IMP2* from human cancer cell lines causes markedly increased abundance of IGFBP2 and Grb14. Extracts prepared from the six cell lines shown in *Figure 1D* were subjected to SDS-PAGE and immunoblot as indicated. (B) Overexpression of IMP2 in human cancer cell lines and MEFs reduces the abundance of IGFBP2 and Grb14. Extracts prepared from the HCC-1419, NCI-H2029 and MEFs stably overexpressing doxycycline-sensitive cDNAs for IMP2, as shown in *Figure 1C* were immunoblotted for IGFBP2 and Grb14. (C) CRISPR mediated deletion of *IMP2* from human cancer cell lines reduces production of IGF2. Aliquots of the medium from the cells depicted in *Figure 1D* were taken at day five and assayed for IGF2; the level in the cell-free medium (approximately 7 ng/ml) has been subtracted. *=p < 0.01 vs parental. (D) Exogenous IGF2 and depletion of Grb14 increases proliferation of IMP2-deficient cancer cells in a preferential and often additive manner as compared with the parental cells. The variants of the Hep3b, HeLa, RD, HCC-1359, MB-231 and SNU-423 cell lines shown in *Figure 1D*, which had been transfected with Cas9/CRISPR and a guide RNA directed at either *GFP* (parental) or *IMP2* (CRISPR) sequences, were infected with Lentiviral vectors encoding a Doxycycline responsive shRNA against *Grb14*. The cells were grown in the presence of DMSO (blue solid line, 'no addition'), DMSO and IGF2 (500 ng/ml, blue dashed line, '+IGF2'), Doxycycline (orange solid line, '+shGrb14'), or Doxycycline and IGF2 (orange dashed line, '+shGrb14 and IGF2'). Cell number was determined daily. The bar graphs show the increase in cell number at day 5 of the cells designated 'CRISPR' divided by the increase in cell number at day 5 of the comparably treated 'parental' cells, multiplied by 100. + = p < 0.05, *=p < 0.01 vs A; x =<0.01 vs B; °=p < 0.05 vs C.
DOI: https://doi.org/10.7554/eLife.27155.006

modest additivity suggesting the contribution of other IMP2-regulated proliferative regulators. Only in the SNU-423 cells (*Figure 4D*, bottom right) did the parental cells show a more robust proliferative response to IGF2 than did the IMP2 deficient variant (2.44 fold vs. 1.63 fold). Nevertheless, IMP2 deficient SNU-423 cells responded to depletion of Grb14 with a greater increase in proliferation than the parental SNU-423 cells (1.5 fold vs 1.12 fold). We conclude that the pro-proliferative

effect of IMP2 in most cancer derived cell lines is mediated predominantly by upregulation of IGF2 production and suppression of Grb14 and IGFBP2 polypeptide abundance which together facilitate IGF2 signaling.

## Reduced HMGA1 underlies the increased abundance of IGFBP2 and Grb14 and the slower proliferation of IMP2 deficient MEFs and tumor cells

Because the increase in *Igfbp2* mRNA induced by IMP2 deficiency is due primarily to increased gene transcription, we inquired whether elimination of IMP2 significantly alters the abundance of any of the IMP2 client RNAs encoding transcription factors (TFs). The RNAseq from *Imp2*$^{+/+}$ and *Imp2*$^{-/-}$ MEFs (*Supplementary file 1*) detected 810 of 1457 mouse TFs at AnimalTFDB (http://www.bioguo.org/AnimalTFDB/index.php), many more than did the proteomic analysis (275/1457); among the 636 RNAs enriched ≥1.5X in the IMP2 IP are 64 that encode TFs, of which 14 show a three-fold or greater decreased abundance in *Imp2*$^{-/-}$ MEFs as compared with *Imp2*$^{+/+}$ MEFs (none show a comparably increased abundance; *Supplementary file 1*). Only one of polypeptides encoded by these 14 mRNAs, HMGA1 was detected in the proteomic analysis (*Supplementary file 2*), and the abundance of HMGA1 polypeptide was approximately 60% lower in the *Imp2*$^{-/-}$ than in the *Imp2*$^{+/+}$ MEFs. We confirmed by PCR that *HMGA1* mRNA is specifically enriched in the IMP2 IP,~2 fold from MEFs and ~5 fold from RD cells (*Figure 5A*); RNA seq indicates that IMP2 binds primarily to the 3'UTR of the *Hmga1* RNA (*Figure 5—figure supplement 1*). *Hmga1* mRNA in *Imp2*$^{-/-}$ MEFs is 90% lower than *Imp2*$^{+/+}$ MEFs (*Figure 5B*; *Supplementary file 1*) primarily because of faster mRNA turnover; the half life of *Hmga1* mRNA, ~3.5 hr in *Imp2*$^{+/+}$ MEFs is reduced to 0.4 hr in *Imp2*$^{-/-}$ MEFs (*Figure 5C*). The reduction in the HMGA1 polypeptide (*Figure 5F*) is somewhat ameliorated by a 1.8 fold increase in *Hmga1* mRNA polysomal abundance (*Figure 5D*).

Depletion of HMGA1 from *Imp2*$^{+/+}$ MEFs or RD cells increased the abundance of *IGFBP2* mRNA 12.1 fold and 20.8 fold respectively (*Figure 5G*, left); reciprocally, doxycycline-induced expression of HMGA1 reduced *IGFBP2* mRNA levels in RD cells and *Imp2*$^{+/+}$ MEFs and by 79% and 63% respectively (*Figure 5G*, right). To determine if HMGA1 directly binds to the mouse *Igfbp2* gene promoter, chromatin immunoprecipitation was performed using MEFs. The *Igfbp2* gene promoter was enriched 3.7 fold in the anti-IMP2 IP as compared with control IgG, an enrichment comparable to that observed of the promoters of the genes encoding *Igf1r, Igfbp1* and *Igfbp3* (*Figure 5H*), previously identified as HMGA1 transcriptional targets (*Aiello et al., 2010*; *Liritano et al., 2012*); notably the promoter of *Grb14* was not enriched in the HMGA1 IP; similar results were obtained with HMGA1 ChIP using RD cells (not shown). Thus HMGA1 binds and regulates the *Igfbp2* gene, suppressing its transcription, so that HMGA1 depletion increases, and overexpression decreases IGFBP2 polypeptide abundance in MEFs; surprisingly, HMGA1 affects the abundance of the Grb14 polypeptide in a virtually identical manner (*Figure 5I*), an effect not mediated through IGFBP2 (*Figure 5—figure supplement 2*). As with loss of IMP2, HMGA1 depletion greatly increases the abundance of Grb14 polypeptide by reducing Grb14 polypeptide degradation (*Figure 5*-Supplementaary *Figure 3*), acting indirectly through an as yet unidentified transcriptional target. The dependence of HMGA1 abundance on IMP2 demonstrated for MEFs (*Figure 5I*) is observed in all of the other cancer derived cell lines examined; elimination of IMP2 is accompanied by a major reduction in HMGA1 polypeptide (*Figure 4A*) whereas forced overexpression of IMP2 in HCC1419 and H2029 cells causes a dose-dependent increase in HMGA1 (*Figure 4B*).

Doxycycline-induced expression of HMGA1 increases the proliferative rate of *Imp2*$^{+/+}$ MEFs by 1.3 fold but increases that of *Imp2*$^{-/-}$ MEFs by 3.1 fold, restoring their proliferative rate from 30.6% of *Imp2*$^{+/+}$ MEFs to 73% (*Figure 5J*), essentially the same increase as incurred by dual depletion of IGFBP2 and Grb14 (*Figure 2D,E*). Induced expression of HMGA1 in RD cells stimulates proliferation by 1.2 fold, whereas HMGA1 increases the proliferative rate of IMP2-deficient RD cells by 2.6 fold, restoring the rate to 83% that of the parental RD cells overexpressing HMGA1 (*Figure 5K*); this is a much greater stimulation than induced by addition of excess IGF2 plus depletion of Grb14 (from 26% to 52% of the parental rate, *Figure 4D* bottom left), suggesting that in RD cells HMGA1 engenders proproliferative outputs in addition its ability to suppress IGFBP2 and Grb14.

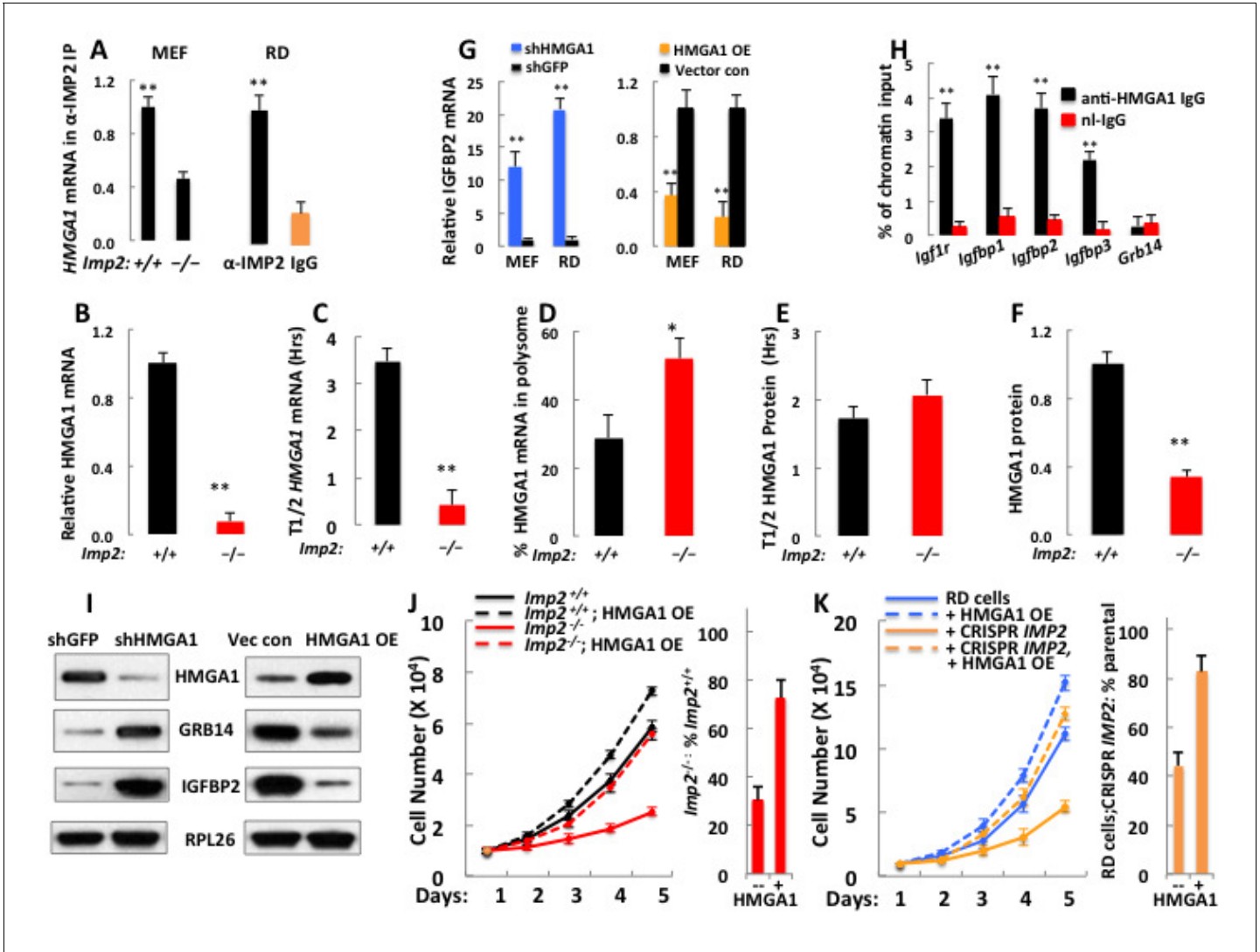

**Figure 5.** A decreased level of *Hmga1*, an IMP2 client mRNA, causes the upregulation of IGFBP2 and Grb14 abundance and the slowed proliferation of IMP2 deficient MEFs and RD cells. (A) IMP2 binds *Hmga1* mRNA. IPs were prepared from extracts of $Imp2^{+/+}$ and $Imp^{-/-}$ MEFs (left pair) and from RD cells (right pair) with anti-IMP2 antibody (black) or nonimmune IgG (orange). The relative enrichment of HMGA1 mRNA in each IP is quantified by real time RT-PCR. (B) *Hmga1* mRNA abundance in $Imp2^{+/+}$ and $Imp2^{-/-}$ MEFs. RNA was extracted from $Imp2^{+/+}$ and $Imp2^{-/-}$ MEFs and the abundance of HMGA1 mRNA was determined by QPCR. The abundance in $Imp2^{-/-}$ MEFs relative to $Imp2^{+/+}$ MEFs (set to 1.0, black bars) is shown in the red bar. (C) The turnover of *Hmga1* mRNAs in $Imp2^{+/+}$ and $Imp2^{-/-}$ MEFs. The t1/2 of *Hmga1* mRNA turnover was determined as in **Figure 3B**. (D) The percentage of *Hmga1*mRNAs residing in polysomes in $Imp2^{+/+}$ and $Imp2^{-/-}$ MEFs. Determined as in **Figure 3C**. (E) The turnover of the HMGA1 polypeptides in $Imp2^{+/+}$ and $Imp2^{-/-}$ MEFs. The t1/2 of HMGA1 polypeptide turnover determined as in **Figure 3D**. (F) The abundance of the HMGA1polypeptides in $Imp2^{+/+}$ and $Imp2^{-/-}$ MEFs. The relative abundance of HMGA1was determined by quantitation of extract immunoblots. Values determined by MS are found in **Supplementary file 2**. For **Figure 5B–F**, +p<0.05, *p<0.01 vs $Imp2^{+/+}$. (G) *IGFBP2* mRNA is regulated by the abundance of HMGA1 polypeptide. *IGFBP2* mRNA in RD cells and $Imp2^{+/+}$ MEFs stably expressing doxycycline-inducible shRNAs against *HMGA1* (left set) or doxycycline-inducible HMGA1 cDNAs (right set). Doxycycline = colored bars; DMSO = black bars. (H) HMGA1 binds the *Igfbp2* but not the *Grb14* promoter. MEFs (~10 million) were washed and protein was cross-linked to DNA using formaldehyde. ChIP was performed as described in Methods using antibodies to HMGA1 (black) or nonimmune IgG (red). (I) Both IGFBP2 and Grb14 protein levels are regulated by HMGA1 polypeptide abundance. Immunoblots were performed using MEFs extracted 72 hr after doxycycline induction of *Hmga1* shRNA (left) or induction of HMGA1 cDNA expression (right). (J) HMGA1 overexpression restores the proliferation of $Imp2^{-/-}$ MEFs close to that of the similarly treated $Imp2^{+/+}$ MEFs. Parental MEFs or MEFs stably expressing a doxycycline-inducible HMGA1 overexpressing MEF cells were treated with doxycycline and cell number was measured daily. The bar graphs were calculated as in **Figure 2E**. (K) The HMGA1 overexpression restores the proliferation of *IMP2* CRISPR RD cells nearly identical to similarly treated parental RD cells. As in **Figure 5J**.

DOI: https://doi.org/10.7554/eLife.27155.007

The following figure supplements are available for figure 5:

*Figure 5 continued on next page*

*Figure 5 continued*

**Figure supplement 1.** RNAseq indicates that IMP2 binds HMGA1 mRNA predominantly at the 3'UTR.

DOI: https://doi.org/10.7554/eLife.27155.008

**Figure supplement 2.** Depletion of IGFBP2 does not affect the abundance of Grb14.

DOI: https://doi.org/10.7554/eLife.27155.009

**Figure supplement 3.** Depletion of HMGA1 prolongs the half-life of the Grb14 polypeptide.

DOI: https://doi.org/10.7554/eLife.27155.010

## IMP2 stimulation of proliferation, stabilization of *Hmga1* mRNA and suppression of Grb14 and IGFBP2 accumulation in MEFs requires an acidic charge at IMP2 residue 164, the mTOR phosphorylation site

We showed previously that transfection of Flag-IMP2(Ser162Asp/Ser164Asp) stimulated the translation of an IGF2-leader 3-luciferase reporter in RD rhabdomyosarcoma cells to an extent similar to Flag-IMP2 wildtype, whereas Flag-IMP2(Ser162Ala/Ser164Ala) was no more effective than empty vector (*Dai et al., 2011*); this translational response was paralleled by the ability of the IMP2 variants to bind the *IMP2 leader 3 5'UTR* RNA. To evaluate the functional significance of each phosphorylation site on the ability of IMP2 to promote proliferation, we generated IMP2 variants with either an Ala or an Asp at both Ser162 and Ser164, which we refer to as IMP2-AA, -AD, -DA and -DD and stably expressed doxycycline regulated Flag-tagged versions of each of the four mutant IMP2, a Flag-tagged IMP2 wildtype (IMP2-SS) and an empty Flag vector in the parental $Imp2^{-/-}$ MEFs. Polyclonal populations were selected and a doxycycline dose was determined that induced polypeptide expression of each Flag-tagged IMP2 variant to a level similar to endogenous IMP2 in $Imp2^{+/+}$ MEFs (*Figure 6A*, immunoblot; *Figure 6B*, TMT peptides, immunoblot below); $10^4$ cells of the six MEF types were plated in replicate and proliferation was monitored over six days (*Figure 6A*). $Imp2^{-/-}$ MEFs expressing IMP2-AD and IMP2-DD proliferated at a rate indistinguishable from $Imp2^{-/-}$ MEFs expressing IMP2-SS, whereas $Imp2^{-/-}$ MEFs expressing IMP2-AA IMP2-DA proliferated at ~50% of that rate as did $Imp2^{-/-}$ MEFs containing empty Flag vector. To evaluate how these IMP2 variants altered the MEF proteome, whole cell mass spectroscopic proteomic analysis was performed as before (*Supplementary file 3*). If the TMT-determined abundance of each polypeptide in the MEFs expressing Flag-IMP2-AA and Flag-IMP2-DA and empty Flag vector (slower proliferating MEFs) were summed, and divided by the sum of the corresponding polypeptide in $Imp2^{-/-}$ MEFs containing Flag-IMP2-SS, Flag-IMP2-DD and Flag-IMP2-AD (faster proliferating MEFs), the two endogenous polypeptides with the highest ratio in the entire proteome are IGFBP2 (5.7) and Grb14 (3.7) (*Figure 6B*, right column). Immunoblot confirmed the ability of IMP2-AD and DD to sustain HMGA1 polypeptide and suppress IGFBP2 and Grb14 polypeptide abundance in $Imp2^{-/-}$ MEFs comparably to wildtype IMP2 and more effectively than IMP2-AA and IMP2-DA (*Figure 6B*, right). Thus the ability of IMP2 to maintain the expression of *Hmga1* and suppress the abundance of IGFBP2 and Grb14, which underlies in large part IMP2's ability to promote MEF proliferation, depends strongly on the status of the IMP2 mTOR phosphorylation site at Ser164.

## IMP2(Ser164) is phosphorylated by mTOR post-translationally as well as co-translationally

The regulation of IMP2 phosphorylation was examined in 293E cells. Overnight withdrawal of serum, brief incubation in amino acid and serum-free medium or torin1 (*Figure 6C*, column 1 vs columns 3,4,5 respectively) fully abolish the concomitant dual phosphorylation at IMP2 Ser162 and Ser164; the sensitivity of this dual phosphorylation to amino acid withdrawal and rapamycin (*Dai et al., 2011*) as well as its rapid stimulation by insulin (*Figure 6C*, column 2 vs column 3) point strongly to the mediation of mTORC1. Nevertheless, serum withdrawal or torin1 cause little dephosphorylation of IMP2(Ser162P) and only partial dephosphorylation of IMP2(Ser164P), a pattern previously observed with shRNA-induced depletion of mTOR in RD cells (*Dai et al., 2011*). The modest dephosphorylation of IMP2(Ser164) caused by torin1 or amino acid withdrawal contrasts with the ability of these treatments to cause the near-total dephosphorylation of sites phosphorylated exclusively by mTORC1, such as S6K1(Ser389) or 4E-BP(Thr37/46). The marked resistance of IMP2(Ser162P) to dephosphorylation by torin1 is similar to that seen previously with the co-translational

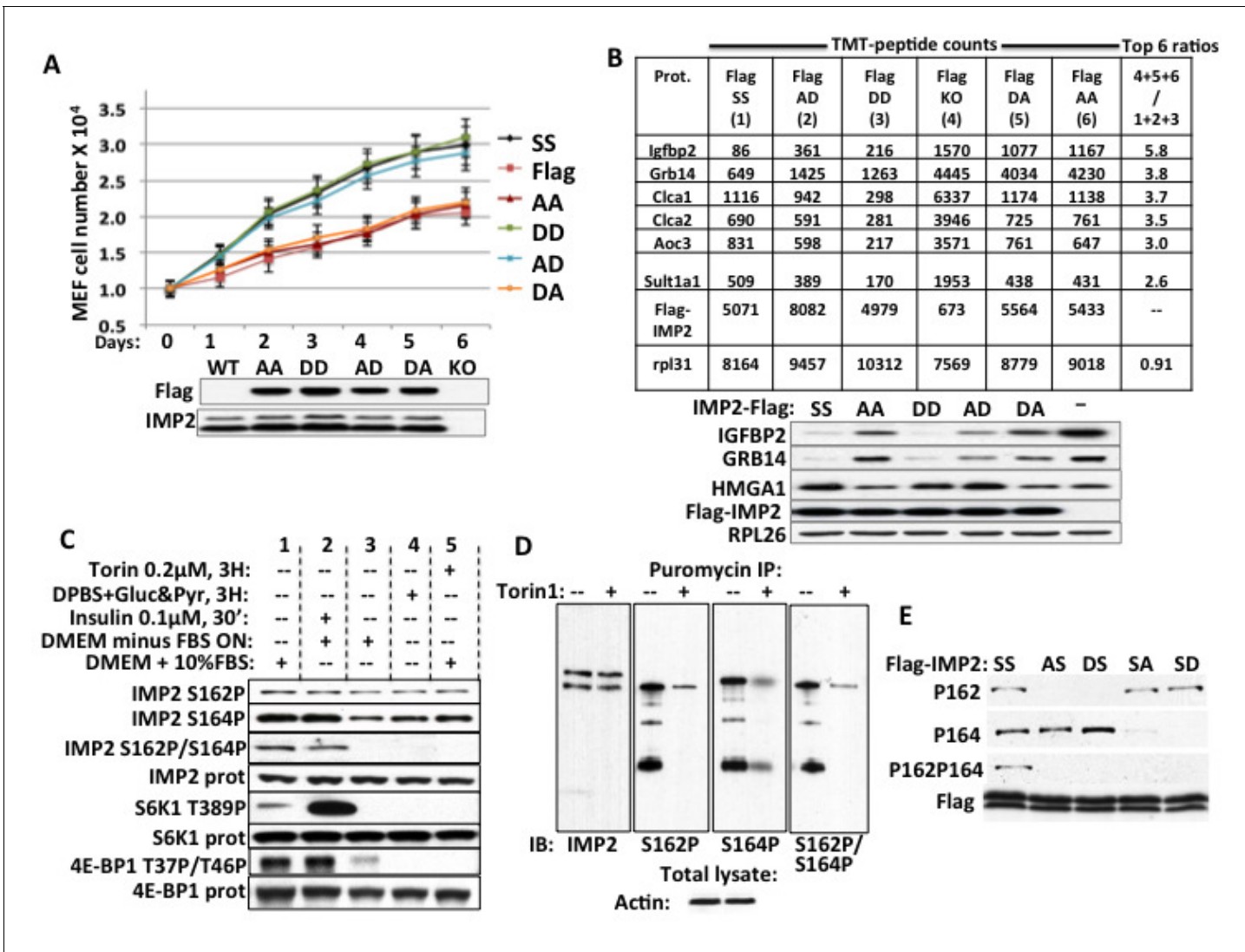

**Figure 6.** The ability of IMP2 to stimulate MEF proliferation and reduce IGFBP2 and Grb14 abundance requires an acidic charge at IMP2 residue 164. (A) The ability of IMP2 to drive MEF proliferation requires an acidic charge at IMP2 residue 164. $Imp2^{-/-}$ MEFs were engineered to express doxycycline-inducible cDNAs encoding Flag tagged IMP2 wildtype (SS), Ser162Ala/Ser164Ala (AA), Ser162Ala/Ser164Asp (AD), Ser162Asp/Ser164Asp (DD), Ser162Asp/Ser164Ala (DA) or empty flag vector (Flag). Doxycycline was adjusted to achieve Flag-IMP2 variant expression approximating the level of IMP2 found in wildtype $Imp2^{+/+}$ MEFs, indicated as WT in the immunoblot. The MEFs expressing the six Flag tagged IMP2 variants were plated in replicate and grown in the presence of the adjusted level of doxycycline. Cell number was determined daily. Each point on the curves for SS, AD and DD do not differ from each other but differ from those for Flag, AA and DA starting at day 2, p<0.01. (B) Genome-wide proteomics reveals that the abundance of the IGFBP2 and Grb14 polypeptides in $Imp2^{-/-}$ MEFs expressing Flag-IMP2 with an Ala at residue 164 is similar to that in MEFs lacking IMP2. Whole cell extracts prepared from the six doxycycline-treated MEFs expressing Flag-IMP2 variants or empty Flag vector shown *Figure 6A* were analyzed by genome wide TMT proteomics. The values for each of the 7964 polypeptides (detected in all six lines) in the (slower growing) Imp2$^{-/-}$ MEFs expressing Flag-IMP2-AA, Flag-IMP2-DA and Flag-vector (KO) were summed and divided by the sum of that polypeptide in MEFs expressing Flag-IMP2 wildtype (SS), Flag-IMP2-AD and Flag-IMP2-DD. This ratio was then sorted from highest to lowest value and the values are shown for the polypeptides showing 6 highest ratios. The abundance of the Flag-IMP2 polypeptide variants and the endogenous rpL31 polypeptide, the latter meant to reflect overall polypeptide abundance, are also shown. The entire data set is in *Supplementary file 3*. Extracts of the MEFs described in *Figure 6A, B* were subjected to SDS-PAGE and immunoblotted as indicated. (C) The phosphorylation of IMP2(Ser164) and concurrent dual phosphorylation of IMP2(Ser162/Ser164) are post-translationally regulated by mTOR. 293E cells were grown continuously in DMEM with 10% FCS (columns 1, 4, 5) were treated with torin1, 200 nM (column 5) or switched to DPBS containing glucose, 25 mM and pyruvate,1mM (column 4) 3 hr prior to extraction some cells. Other 293 cells were subjected to overnight serum withdrawal (columns 2,3) followed by addition of insulin 100 nM for 30 min. (column 2) prior to extraction. The extracts were subjected to SDS-PAGE and immunoblot as indicated. (D) Nascent IMP2 polypeptides bound to puromycin are co-translationally phosphorylated at Ser162 and Ser164 by mTOR. Twenty min. after addition of torin1 (200 nM) to rapidly growing RD cells, puromycin was added at 10 ug/ml, and the cells were extracted 10 min. later. The extracts were blotted for IMP2 and puromycin immunoprecipitates were blotted for IMP2 and IMP2(Ser162P) and IMP2(Ser164P). (E) The modification of Ser162 does not affect the phosphorylation of Ser164 and vice versa. 293 cells were

*Figure 6 continued on next page*

*Figure 6 continued*

transiently transfected with vectors encoding Flag-tagged wildtype IMP2 (SS) or the Flag-IMP2 mutants Ser162Ala (AS), Ser162Asp (DS), Ser164Ala (SA) or Ser164Asp (SD). Flag immunoprecipitates were separated by SDS-PAGE and immunoblotted as indicated.

DOI: https://doi.org/10.7554/eLife.27155.011

phosphorylation of IMP1(Ser181) (*Dai et al., 2013*). We therefore examined whether IMP2 is phosphorylated co-translationally by incubating 293 cells with puromycin, which is covalently incorporated into nascent polypeptides because of its resemblance to the 3' end of aminoacylated tRNA, thereby terminating elongation. Puromycin immunoprecipitates contain IMP2 polypeptides that are phosphorylated at both Ser162 and Ser164 as well as at both sites concomitantly, which can have only occurred co-translationally; pretreatment of the cells with torin1 20 min. prior to addition of puromycin strongly inhibits these phosphorylations of nascent IMP2 (*Figure 6D*). Thus IMP2 undergoes mTOR dependent cotranslational phosphorylation at both Ser162 and Ser164, in addition to an insulin-, serum- and rapamycin-sensitive post-translational phosphorylation primarily (if not exclusively) at IMP2(Ser164), which results in concomitant dual phosphorylation. How the mature IMP2 polypeptide attains its steady state level of phosphorylation is not entirely understood; it is clear however, that the phosphorylation at either IMP2 site does not affect phosphorylation at the other site; substitution of Ser164 with either Ala or Asp has little effect on the extent of phosphorylation at Ser162 and the same is true of such substitutions at Ser162 on the phosphorylation at Ser164 (*Figure 6E*).

## Discussion

*IMP2* is amplified and overexpressed in many cancers and recent reports (*Li et al., 2015*; *Liu et al., 2015*; *Mu et al., 2015*) have pointed to a tumorogenic role in individual cancer types, often with a poor prognosis (*Barghash et al., 2015*; *Bigagli et al., 2016*; *Davidson et al., 2014*; *Kessler et al., 2013*). Consistent with the view that IMP2 contributes broadly to tumor progression, we show herein that IMP2 drives the proliferation of both MEFs and a wide and unselected cohort of cancer cell lines. IMP2 is best considered a tumor promoter rather than an oncogene; although mice with transgenic overexpression of IMP2 in liver exhibit a higher tumor burden after diethylnitrosamine-induced hepatotoxicity (*Kessler et al., 2015*), IMP2 has not as yet been shown capable of initiating tumorigenesis. The conclusion that IMP2 is a tumor promoter raises for consideration the biology of IMP2 in normal, noncancerous circumstances and how this might underlie tumor promotion. *IMP1* and *IMP3* are so-called 'oncofetal' genes because their high expression in embryonic development is strongly downregulated before birth, but frequently reappears in cancers. *IMP2* diverges from this model in that it continues to be expressed widely in adult life, however IMP2 appears to be preferentially expressed in tissue resident stem cells, such as satellite cells (*Boudoukha et al., 2010*; *Li et al., 2012*), neural stem cells (*Fujii et al., 2013*) and white adipocyte precursors (*Dai et al., 2015*), where it is critical for their ability to proliferate, differentiate and gain motility. Thus *IMP2* gene amplification and/or upregulated expression may be important to the genesis of cancer stem cells and thereby central to IMP2's tumor promoting capability, as proposed by (*Degrauwe et al., 2016a*) and demonstrated in glioblastoma stem cells (*Degrauwe et al., 2016b*).

The present work also elucidates a major molecular pathway that underlies IMP2's tumor promoting activity; IMP2 occupies a central place in a network of gene products that drive proliferation through the IGF1R and INSR-A (*Figure 7*). The expression of the *IMP2* gene is strongly upregulated by the oncofetal HMGA2 oncogene (*Brants et al., 2004*; *Cleynen et al., 2007*; *Li et al., 2012*) and the mRNA encoding the oncogenic transcriptional modifier *HMGA1* (*Fedele and Fusco, 2010*; *Ozturk et al., 2014*, *Sumter et al., 2016*), previously detected as an IMP2 client (*Janiszewska et al., 2012*), is identified here as an important mediator of IMP2's tumor promoting activity. Restoring HMGA1 abundance largely rescues the proliferative defect engendered by IMP2 deficiency in MEFs and RD cells (*Figure 5J,K*). IMP2 binds the *HMGA1* mRNA 3'UTR, inhibiting its degradation; elimination of IMP2 causes a major decrease in *HMGA1* mRNA and protein abundance. Regarding mechanism by which IMP2 protects *HMGA1* mRNA, *Degrauwe et al. (2016b)* demonstrated in Glioblastoma stem cells that IMP2 binding to the *HMGA1* mRNA 3'UTR (*Figure 5—figure*

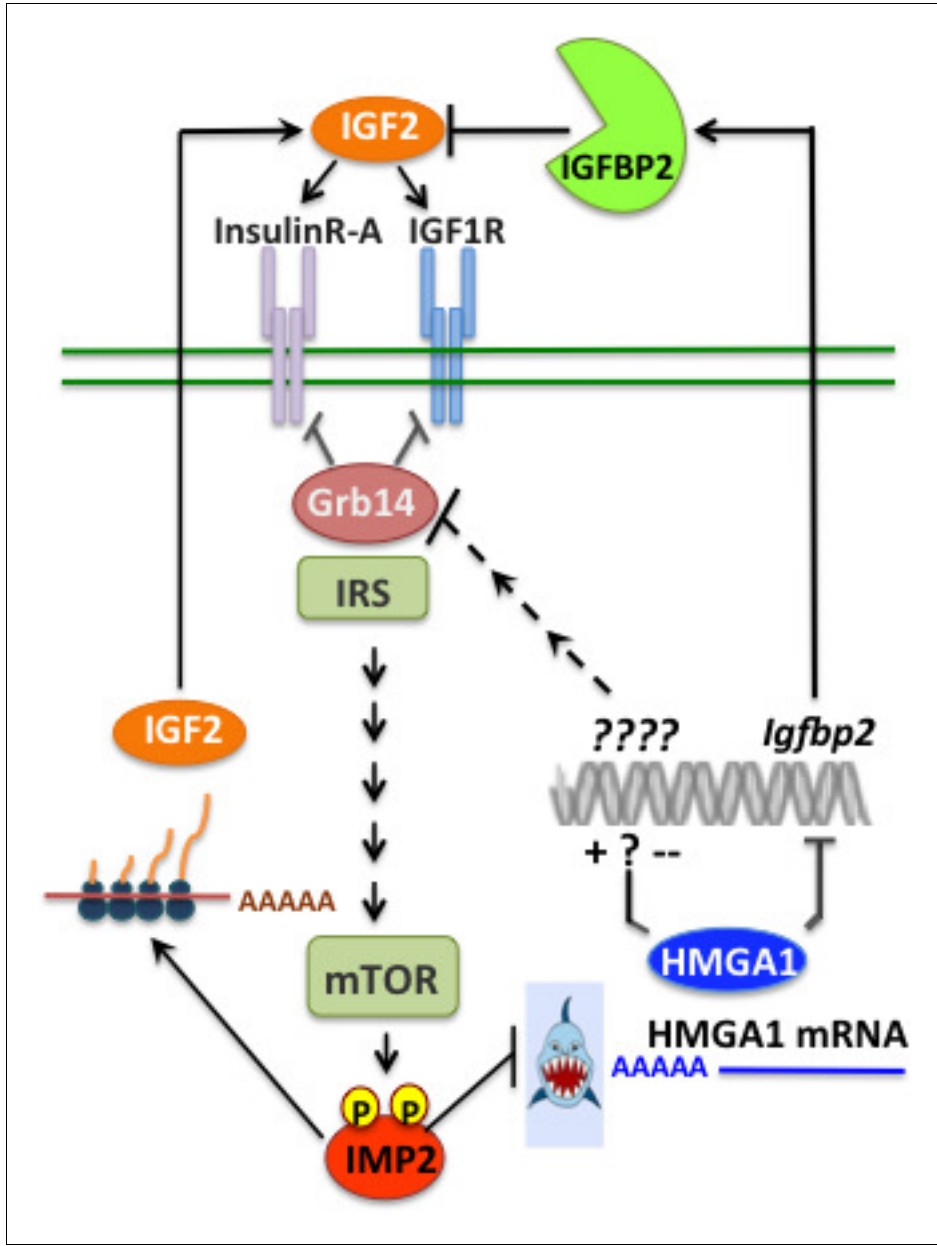

**Figure 7.** IMP2 promotes proliferation by increasing IGF2 polypeptide and by stabilizing *HMGA1* mRNA, whose polypeptide product suppresses the abundance of IGFBP2 and Grb14, inhibitors of IGF2 action.
DOI: https://doi.org/10.7554/eLife.27155.012

supplement 1) overlaps the binding site of the pro-differentiation tumor suppressor miR, *Let-7* and antagonizes *Let-7*'s ability to promote *HMGA1* mRNA degradation.

*HMGA1* expression is high during embryonic development, much lower in adult tissues, but greatly upregulated in a broad range of benign and malignant tumors. The HMGA1 protein binds to the AT-rich regions in the DNA minor groove; although lacking direct transcriptional regulatory activity, it modulates gene expression by changing chromatin structure and promoting the assembly of other TFs. HMGA1's output is thus cell context dependent; in addition to oncogenic transformation, HMGA1 can also promote expression of a stem cell-like program and/or an epithelial-to-mesenchymal transition. Numerous transcriptional targets relevant to tumorigenesis have been described, including *p53* and *Rb* (*Sumter et al., 2016*) or the oncogenic miR222 (*Wong et al., 2010*; *Zhang et al., 2011*; *Panneerselvam et al., 2016*); here we identify the ability of HMGA1 to

suppress the abundance of two previously unknown targets, IGFBP2 and Grb14, inhibitors of IGF1R and Insulin receptor signaling. HMGA1 binds directly within the *Igfbp2* gene promoter to suppress transcription. In contrast, HMGA1 does not bind to the *Grb14* gene, but acts through an unidentified transcriptional target to ultimately promote Grb14 polypeptide degradation.

Envisioning an anti-proliferative role for Grb14 is relatively straightforward, although contrary examples exist (*Balogh et al., 2012*). The role of IGFBP2 in cancer however is controversial (*Pickard and McCance, 2015*; *Russo et al., 2015*). IGFBP2 has IGF-dependent and independent, proproliferative actions in cell culture; IGFBP2 contains an RGD domain and can interact with the integrins αVβ3 and α5β1 to affect cell survival or motility (see *Russo et al., 2015*), and functions coordinately with IGF1 to promote endothelial proliferation through polymerization and inactivation of the receptor tyrosine phosphatase RTPTβ, causing inhibition of PTEN (*Shen et al., 2015*). Nevertheless, the ability of transgenic overexpression of IGFBP2 to inhibit postnatal growth (*Hoeflich et al., 1999*) and colonic tumorigenesis (*Diehl et al., 2009*) attests to its ability to sequester IGF in vivo.

A central role for IGFs in tumorigenesis had been predicted by the finding that mouse embryo fibroblasts devoid of an IGF1R are resistant to transformation by many cellular and viral oncogenes (*Sell et al., 1993*). Although studies in mouse cancer models were often supportive, interventions directed at inhibition of the IGF1R in man as a single therapy and in combination with chemotherapy have not shown efficacy (*Baserga, 2013*; *Pollak, 2012*). The reason(s) for this disappointing outcome are not known, however one likely mechanism is the frequent overexpression of IGF2, due in part to loss of imprinting at the IGF2 locus and/or loss of the IGF2R (*Brouwer-Visser and Huang, 2015*; *Livingstone, 2013*). IGF2 has high affinity for the fetal-predominant A form of the insulin receptor (INSR-A), also frequently reexpressed in tumors. The present findings identify IMP2 as a vital component of this network; its stabilization of *HMGA1* mRNA is necessary for the oncogene-driven upregulation of *HMGA1* transcription to generate HMGA1 polypeptide overabundance. In turn, HMGA1 suppression of Grb14 and IGFBP2 synergize with IMP2 stimulation of *IGF2 L3* mRNA translation to promote IGF2 mitogenic signaling though the IGF1R and Insulin Receptor-A (*Morrione et al., 1997*; *Ulanet et al., 2010*).

It is worth noting that the human genes encoding major elements of the tumor promoting pathway described herein, that is, *HMGA2, IMP2, Grb14, IGF2*, have each been found by GWAS to encode SNPs that are associated with excess risk for type 2 diabetes (*Flannick and Florez, 2016*), a condition accompanied by increased risk for many cancers (*Giovannucci et al., 2010*; *García-Jiménez et al., 2016*). Most of these SNPs are noncoding so that the contribution of the encoded polypeptide in conferring risk is unknown. As regards HMGA1, *Foti et al. (2005)* reported 4 individuals with reduced HMGA1 abundance due to either hemizygous gene loss or point mutation, who exhibited severe insulin resistance accompanied by diminished insulin receptor abundance. This group also described variants in the *HMGA1* gene that associate in excess with type 2 diabetes (*Chiefari et al., 2011*) however this association has been effectively refuted (*Marquez et al., 2012*; *Froguel et al., 2012*), so that a role for HMGA1 in human type 2 diabetes remains unsubstantiated. The mouse HMGA1 KO also exhibits reduced insulin receptor abundance but as part of a complex metabolic phenotype with mild glucose intolerance, hypoinsulinemia and enhanced insulin sensitivity (*Foti et al., 2005*). Thus, whereas the ability of the pathway centered around IMP2 to initiate and/or promote tumor progression is well established, the role of these elements, if any, in the development of type 2 diabetes and the increased susceptibility of that condition to various cancers remains to be elucidated.

## Materials and methods

### Cell culture

MEFs were isolated from E12.5–13.5 embryos of Imp2$^{+/-}$ intercrosses (*Dai et al., 2015*). The proliferation, QPCR and polypeptide abundance measurements derived from mouse embryo fibroblasts (*Figure 1E,F,G Figure 2D,E*, *Figure 3A–E*, *Figure 5A–H,J*, *Figure 6A*) represent data from 3 or more separate experiments; each experiment used MEFs derived from different Imp2$^{+/+}$ and Imp2$^{-/-}$ embryos. Human RD (RRID:CVCL_1649), Hela (RRID:CVCL_0058), MB-231 (RRID:CVCL_0062), Hep3b (RRID:CVCL_0326) and 293E (RRID:CVCL_6974) cell lines were maintained in DMEM

supplemented with 10% FBS. HCC-1359 (RRID:CVCL_5128), NCI-H2029 (RRID:CVCL_1516), SNU-423 (RRID:CVCL_0366), HCC-1419 (RRID:CVCL_1251) cells were cultured in RPMI-1640 supplemented with 10% FBS. The corresponding data derived from human cancer cell lines (*Figure 1C,D*, *Figure 4C,D*, *Figure 5A,G,K*) represented three separate experiments. The error bars on these figures represent ± one S.D.

## Reagents and antibodies

DMSO, doxycycline, cycloheximide, rapamycin, insulin, human IGF1, human IGF2, RPL26 and actin antibodies were from Sigma-Aldrich (St. Louis, Missouri, USA); Torin 1 from TOCRIS. Antibodies: P-AKT(Ser473), AKT, pS6K(Thr389), S6K, P-4E-BP1(Thr37/460), 4E-BP1, P-AMPK, AMPK, IGF1R, IRS1, IRS2, P21, Phos-Tyrosine from Cell Signaling Technology; Grb14 and IGFBP2 antibodies from abcam. IMP antibodies were described previously (*Dai et al., 2011*; *Dai et al., 2013*). The inducible Lentiviral shRNA for *Grb14, IGFBP2, p21* and *HMGA1* were from Dharmacon and the stably expressed Flag-IMP2 variants and Flag-HMGA1were generated using pcDNA6/TR and pcDNA5/TO vectors; these reagents were used according to manufacturer's instructions. The P-site mutants of IMP2 were generated using the QuikChange site-directed mutagenesis kit (Stratagene).

## hIMP2 CRISPR

CRISPR-Cas reagents were obtained from Addgene (U6-Chimeric_BB-CBh-hSpCas9). Small guide RNAs (sgRNAs), selected for minimal predicted off-target mutagenesis, were designed using CRISPR Design software (http://crispr.mit.edu); (hIMP2_CRISPR 1, caccgACAAGAACAATTCCTGAGCT; hIMP2_CRISPR 2, aaacAGCTCAGGAATTGTTCTTGTC). Plasmids co-expressing the sgRNA specific to the *IMP2* and the *Streptococcus pyogenes* Cas9 nuclease were transfected into cells by Nucleofector (Lonza) according to manufacturer's instruction.

## Protein analyses

Immunoblot analyses as described (*Dai et al., 2015*); each immunoblot was repeated at least one time. The levels of IGF2 were determined using Human IGF2 and Mouse IGF2 ELISA kit from Cusabio life science and R&D systems respectively. Cycloheximide was used at a final concentration of 250 μM to inhibit mRNA translation; protein half-life was estimated from immunoblot of extracts prepared before and at various times after cycloheximide addition.

## Analysis of human cancer genomic data

The data used to construct the comparison of *IGF2BP1/2/3* gene copy number (*Figure 1A*) and mRNA abundance (*Figure 1B*) in human cancers was obtained from TCGA published (PMID:18872890,21720365,20579941,20601955,27158780,28112728, 28052061,26061751) and provisional datasets. cBioPortal (http://www.cbioportal.org/index.do) was used to organize and process the TCGA data.

## Multiplexed quantitative mass spectrometry analysis

Samples were processed and analyzed through the Thermo Fisher Scientific Center for Multiplexed Proteomics at Harvard Medical School (*Weekes et al., 2014*). Peptides derived from digestion using LysC and trypsin were labeled with Tandem Mass Tag 8-plex reagents and fractionated. Multiplexed quantitative mass spectrometry data were collected on an Orbitrap Fusion mass spectrometer operating in a MS3 mode using synchronous precursor selection for the MS2 to MS3 fragmentation (*McAlister et al., 2014*). MS/MS data were searched against a Uniprot mouse database with both the forward and reverse sequences using the SEQUEST algorithm. Further data processing steps included controlling peptide and protein level false discovery rates, assembling proteins from peptides, and protein quantification from peptides.

## RNA analysis

RNA was isolated using QIAGEN RNase kit. RNA samples were examined for the integrity by Agilent bioanalyzer prior to further processing. Sequencing was carried out on Illumina HiSeq 2500, resulting in approximately 21 million of 50 bp paired-end reads per sample. STAR aligner (Pubmed ID: 23104886) was used to map sequencing reads to the Mus musculus genome version 10 (mm 10)

reference genomes. Read counts for individual transcripts were produced with HTSeq-count (Pubmed ID:25260700), followed by the estimation of expression values and detection of differentially expressed transcripts using EdgeR (Pubmed ID:19910308). The RNA sequence data have been deposited in NCBI's Gene Expression Omnibus and are accessible through GEO series accession number GSE101311.

## mRNAs half-lives determination

Cells were treated with 1 µM Actinomycin D (Sigma) for 12 hr to block transcription. After the indicated times, the total RNAs were extracted, followed by DNase digestion for eliminating DNA contamination and cDNA syntheses. The concentration of mRNAs were quantified by Real-Time RT-PCR using the SYBR Green. The primers for the real-time PCRs were as follow: murine *Grb14* sense (actatgtggacgacaacagc), antisense (cataattctttctaagatacag); murine *Igfbp2* sense (ccagcaggagttggaccaggt), antisense (cttaaggttgtaccggccatgc); murine *P21*sense (cgagaacggtggaactttgac), antisense (cagggctcaggtagaccttg).

## Chromatin immunoprecipitation (ChIP)

Protein was cross-linked to DNA by treatment of PBS washed cells with formaldehyde. Cells were pelleted and resuspended in cell lysis buffer containing a protease inhibitor cocktail (Roche). After 10 min on ice, the nuclei were collected and chromatin was sheared by sonication. ChIP was performed using the Simple ChIP Enzymatic Chromatin IP kit (Cell Signaling Technology) according to the manufacturers' instructions. The sheared DNA-protein complexes were immune-precipitated using antibodies to HMGA1. A nonimmune IgG was the negative control.

## Acknowledgements

ND and JA designed and supervised the studies, JW and LM provided reagents, ND and FJ acquired the data, ND, FJ, RS and JA analyzed the data and contributed to writing and all au reviewed and approved the final draft. We thank J Florez for discussions. Supported by NIH awards DK017776, DK057521, DK040561 and institutional sources.

## Additional information

### Funding

| Funder | Grant reference number | Author |
|---|---|---|
| National Institute of Diabetes and Digestive and Kidney Diseases | R37 DK17776 | Joseph Avruch |
| National Institute of Diabetes and Digestive and Kidney Diseases | P30 DK057521 | Joseph Avruch |
| National Institute of Diabetes and Digestive and Kidney Diseases | P30 DK040561 | Ruslan Sadreyev |

The funders had no role in study design, data collection and interpretation, or the decision to submit the work for publication.

### Author contributions

Ning Dai, Conceptualization, Investigation, Writing—original draft, Writing—review and editing, Designed and supervised the studies, acquired the data, analyzed the data, contributed to writing, and reviewed and approved the final draft; Fei Ji, Data curation, Formal analysis, Investigation, Writing—review and editing, Acquired the data, analyzed the data, contributed to writing, and reviewed and approved the final draft; Jason Wright, Resources, Methodology, Provided reagents, reviewed and approved the final draft; Liliana Minichiello, Resources, Writing—review and editing, Provided reagents and reviewed and approved the final draft; Ruslan Sadreyev, Supervision, Methodology,

Writing—review and editing, Analyzed the data, contributed to writing and reviewed and approved the final draft; Joseph Avruch, Conceptualization, Supervision, Funding acquisition, Writing—original draft, Writing—review and editing, Designed and supervised the studies, analyzed the data, contributed to writing, and reviewed and approved the final draft

**Author ORCIDs**
Joseph Avruch https://orcid.org/0000-0003-4940-3495

**Decision letter and Author response**
Decision letter https://doi.org/10.7554/eLife.27155.016
Author response https://doi.org/10.7554/eLife.27155.017

## Additional files

**Supplementary files**
• Supplementary file 1. RNA seq of total RNA and of IMP2 immunoprecipitates from $Imp2^{+/+}$ and $Imp2^{-/-}$ MEFs.
DOI: https://doi.org/10.7554/eLife.27155.013

• Supplementary file 2. Proteomic analysis of $Imp2^{+/+}$ and $Imp2^{-/-}$ MEFs left tab = summary middle tab = all data right tab = polypeptide ratios: $Imp2^{-/-}/Imp2^{+/+}$.
DOI: https://doi.org/10.7554/eLife.27155.014

• Supplementary file 3. Proteomic analysis of $Imp2^{-/-}$ MEFs stably expressing empty Flag vector (designated C) or Flag-IMP2, wildtype (Ser162/Ser164) or with Ser162 and Ser164 substituted with either Ala or Asp: WT/AD/DD/DA/AA/.
DOI: https://doi.org/10.7554/eLife.27155.015

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
