## [Decision Letter]

Thank you for submitting your article "IGF2 mRNA binding protein-2 is a tumor promoter that drives cancer proliferation through its client mRNAs IGF2 and HMGA1" for consideration by *eLife*. Your article has been reviewed by two peer reviewers, and the evaluation has been overseen by a Reviewing Editor and Charles Sawyers as the Senior Editor. The reviewers have opted to remain anonymous.

The reviewers have discussed the reviews with one another and the Reviewing Editor has drafted this decision to help you prepare a revised submission.

Summary:

The report addresses potential mechanisms whereby the IGF-2 mRNA binding protein IMP2 promotes tumor growth and progression. The findings reveal new information regarding the role of IMP2 in regulating HMGA1 intracellular concentrations to modulate IGFBP-2, IGF-II and Grb14. The study also elucidates the role of these factors in regulating tumor cell proliferation. Extensive genomic and proteomic analyses clearly support the conclusion that these are the predominant factors that act together to regulate this signaling system. Overall, the findings are clearly presented and conclusions are well justified by the experimental data.

Essential revisions:

1) One might expect there to be a synergy between IGFBP2 and Grb14, but Figure 2 suggests that there isn't one. Perhaps the investigators could include this in the discussion, as it would suggest that in their model IGFBP2 inhibits proliferation primarily by sequestering IGF2. This actually strengthens their hypothesis.

2) Figure 2: there appears to be constitutive phosphorylation of Akt in IMP2^+/+^ cells, which does not appear to be significantly altered (judging by the blots) in response to IGF1. Moreover, whereas the robust decrease in IGF-1R and IRS phosphorylation in IMP2^-/-^ cells is partially rescued upon addition of IGF1, rescue of the corresponding decrease in Akt phosphorylation appears to be minimal. This discrepancy should be explained.

3) The mechanism of Grb14 protein stabilization by IMP2 is unexplained. Given that Grb14 plays a role that is shown to be as important as that of IGFBP2 in disrupting IGF2-mediated cell growth, some insight into the mechanism whereby IMP2 stabilizes Grb14 ought to be provided. For example, does IGFBP2 exert an effect, direct or indirect, on Grb14 expression? Does depletion of IGFBP2 alter Grb14 expression? And if HMGA1 does not affect Grb14 stability, what are the candidate factors that may do so?

4) The mechanism of HMGA1 mRNA protection by IMP2 is unexplored. Without going into excessive depth here, the investigators could easily determine whether IMP2 binds to let-7 binding sites in the 3' UTR of HMGA1. This would at least suggest whether or not HMGA-1 mRNA stability is mediated by the ability of IMP2 to block or displace let-7 family miRNA binding to their targets.

5) The investigators begin their discussion by stating that their findings define the functional significance of IMP2 in cancer cells. However, other observations have shown that IMP2, in addition to other IMPs, promotes stemness in cancer cells. There are therefore probably several, possibly synergistic, mechanisms whereby IMP2 promotes tumor growth and this notion should be included in the Discussion. Furthermore, the investigators cite several studies that show a correlation between the poor prognosis of a variety of cancers and IMP2 expression. Recent work (Degrauwe et al., as well as others) has revealed precise mechanisms whereby IMP2 may be implicated in tumor maintenance and growth and should be cited.

6) The results regarding P21 are interesting however it is not clear why lowering P21 levels had no effect on proliferation. The assumption would be that even after deliberately lowering the levels they are still relatively high and therefore even a 5.9 fold change in levels is not adequate to alter proliferation. Some comment should be made on these findings. The authors comment in two places in the manuscript that the changes in proliferation could be in part mediated through insulin receptor A. Since Foti et al. (Nature Medicine 2005) reported that knocking out HMG A1 markedly downregulated insulin receptor expression and that transfection of the cells with HMG A1 restored insulin receptor expression one would expect the manipulations herein to alter insulin receptor number. If these manipulations of HMGA1 do change insulin receptor expression then tumor proliferation could be driven through this receptor. Therefore, showing the change in insulin receptor number would strongly support their hypothesis that this is the mechanism mediating tumor cell proliferation under these conditions. It seems reasonable to measure insulin receptor expression under conditions of high and low HMGA1 expression.

7) The authors should discuss the data of Foti et al. and how they apply to their system. Additionally, those authors (Liritano et al. Mol endo 2012) have published data indicating that with HMG A1 downregulation there is enhanced insulin/IGF-I sensitivity which, if discussed, would also contribute to a more complete consideration of their hypothesis.

---

## [Author Response]

Essential revisions:

1) One might expect there to be a synergy between IGFBP2 and Grb14, but Figure 2 suggests that there isn't one. Perhaps the investigators could include this in the discussion, as it would suggest that in their model IGFBP2 inhibits proliferation primarily by sequestering IGF2. This actually strengthens their hypothesis.

Depleting Grb14 or IGFBP2 from Imp2^-/-^ MEFs each restores the proliferative rate from ~25% to ~70-80% of similarly treated Imp2^+/+^ MEFs and dual depletion is non-additive. We agree with the reviewer that this non-additivity suggests that both interventions act through the same pathway, i.e., by restoring signaling though the InsR and/or IGF-1R. We have added a sentence to that effect (subsection “Depletion of Grb14 and IGFBP2 each partially rescue Imp2^-/-^ MEF proliferation but depletion of p21Cip1 does not” last paragraph).

The non-additivity and known ability of IGFBP2 to bind IGFs certainly supports the view that the predominant action of overabundant IGFBP2 in the Imp2^-/-^ MEFs is to sequester autocrine IGF, but doesn’t rule out other possible sites of IGFBP2 action on that pathway.

The finding that neither intervention enables complete restoration of Imp2^-/-^ MEFs proliferation and that together they are not additive may indicate that each intervention is sufficient to regain the full effect of the InsR/IGF1R pathway on the proliferation of Imp2^-/-^ MEFs; i.e., impaired signaling by the InsR/IGF1R pathway perhaps accounts for ~2/3 of the ~75% reduced proliferation of Imp2^-/-^ MEFs and other, as yet unidentified IMP2-dependent factors, that act through other proproliferative pathways or processes, account for the remaining deficit.

2) Figure 2: there appears to be constitutive phosphorylation of Akt in IMP2^+/+^ cells, which does not appear to be significantly altered (judging by the blots) in response to IGF1. Moreover, whereas the robust decrease in IGF-1R and IRS phosphorylation in IMP2^-/-^ cells is partially rescued upon addition of IGF1, rescue of the corresponding decrease in Akt phosphorylation appears to be minimal. This discrepancy should be explained.

We’ve carried out this experiment three times and the other 2 experiments are shown in Author response image 1. In the Imp2^+/+^ MEFs, basal Akt(Ser473P) is high and IGF1 causes a somewhat variable increase in Akt(Ser473P). The striking and consistent finding is the markedly lower basal and IGF1-stimulated Akt(Ser473P) signal in the Imp2^-/-^ MEFs. Nevertheless, IGF1 addition does increase the Akt(Ser473P) signal in the Imp2^-/-^ MEFs in all three experiments. Regardless of the difference in the basal activity of Akt, we think that the lesser Tyr-P of the IGF1R and IRS polypeptides in the Imp2^-/-^ MEFs is strongly indicative of impaired IGF signaling.

**Author response image 1. respfig1:** IGF1R signaling is impaired in Imp2^-/-^ MEFs.

We have not pursued experimentally the mechanism underlying the difference in the basal/stimulated activation state of Akt in Imp2^+/+^ vs. Imp2^-/-^ MEFs. HMGA1 has been reported to suppress the activity of PP2A, by driving expression of miR222, which promotes degradation of the mRNA encoding PPP2R2A (PP2A subunit Bα), reducing PP2A activity and activating Akt(Wong et.al., 2010, Zhang et.al., 2011, Paneerselvam et.al., 2016). Our proteomic analysis (Supplementary file 2) however finds similar levels of PPP2R2A in Imp2^+/+^ and Imp2^-/-^ MEFs. Thus, we have no satisfactory explanation for the difference in Akt activity in serum-deprived Imp2^+/+^ and Imp2^-/-^ MEFs.

3) The mechanism of Grb14 protein stabilization by IMP2 is unexplained. Given that Grb14 plays a role that is shown to be as important as that of IGFBP2 in disrupting IGF2-mediated cell growth, some insight into the mechanism whereby IMP2 stabilizes Grb14 ought to be provided. For example, does IGFBP2 exert an effect, direct or indirect, on Grb14 expression? Does depletion of IGFBP2 alter Grb14 expression? And if HMGA1 does not affect Grb14 stability, what are the candidate factors that may do so?

We have been unable to identify the proximate mechanism by which IMP2 regulates the degradation of Grb14. Clearly, HMGA1 (Figure 4), like IMP2 (Figure 2; Figure 3), regulates Grb14 abundance, and as with reduced IMP2, the effect of reduced HMGA1 is to prolong the half-life of the Grb14 polypeptide (Figure 5—figure supplement 3); this change is not mediated by IGFBP2 (Figure 5—figure supplement 2). These data are described in the subsection “Reduced HMGA1 underlies the increased abundance of IGFBP2 and Grb14 and the slower proliferation of IMP2 deficient MEFs and tumor cells”.

We have therefore hypothesized that IMP2 acts through HMGA1 which regulates the abundance of a E2 or E3 component of the ubiquitination pathway, a deubiquitinase, a protease or protease inhibitor, or some gene product (including miRs) that regulate the abundance of such elements. The number of steps that intervene between HMGA1 and Grb14 degradation is unknown. We have been unable to identify plausible candidates among the HMGA1 transcriptome and have not identified the elements that directly control Grb14 degradation.

4) The mechanism of HMGA1 mRNA protection by IMP2 is unexplored. Without going into excessive depth here, the investigators could easily determine whether IMP2 binds to let-7 binding sites in the 3' UTR of HMGA1. This would at least suggest whether or not HMGA-1 mRNA stability is mediated by the ability of IMP2 to block or displace let-7 family miRNA binding to their targets.

RNAseq of IMP2 IPs confirms that IMP2 binds extensively to the HMGA1 mRNA 3’UTR (Figure 5—figure supplement 1), and as shown by Degrauwe et.al., (2016b), IMP2 overlaps the Let-7 MRE in the HMGA1 3’UTR. Those workers demonstrated convincingly that IMP2 antagonizes Let-7 mediated HMGA1 mRNA degradation. We cite this work in the second paragraph of the Discussion, as providing a likely mechanism for the marked reduction in HMGA1 mRNA we observe with IMP2 deletion; we have not attempted to repeat those experiments.

5) The investigators begin their discussion by stating that their findings define the functional significance of IMP2 in cancer cells. However, other observations have shown that IMP2, in addition to other IMPs, promotes stemness in cancer cells. There are therefore probably several, possibly synergistic, mechanisms whereby IMP2 promotes tumor growth and this notion should be included in the Discussion. Furthermore, the investigators cite several studies that show a correlation between the poor prognosis of a variety of cancers and IMP2 expression. Recent work (Degrauwe et al., as well as others) has revealed precise mechanisms whereby IMP2 may be implicated in tumor maintenance and growth and should be cited.

This is an important point, and we have modified the opening paragraph of the Discussion, to emphasize this aspect of IMP2 biology. We stress that in adult life IMP2, in contrast to IMP1/IMP3, is widely expressed and appears to be preferentially expressed in tissue resident stem cells, citing examples where IMP2 has been shown to be important to their ability to both proliferate and differentiate. We then cite the work of Degrauwe et.al., 2016b, which demonstrates the importance of IMP2 in glioblastoma stem cells.

6) The results regarding P21 are interesting however it is not clear why lowering P21 levels had no effect on proliferation. The assumption would be that even after deliberately lowering the levels they are still relatively high and therefore even a 5.9 fold change in levels is not adequate to alter proliferation. Some comment should be made on these findings.

Although P21 CDKi is ~5X more abundant in IMP2^-/-^ MEFs than in IMP2^+/+^ MEFs, depletion of P21 to nearly the same level accelerates proliferation by only ~30% in both cell types. This similarity suggests to us that the impedance of the IGF1R/InsR pathway in the IMP2^-/-^ MEFs imposes a restraint on proliferation that is largely independent of that imposed by P21.That said however, we have no compelling explanation for why depletion of P21 accelerates proliferation to a similar extent in WT and IMP2 KO MEFs. An obvious possibility is that the restraint to cell proliferation attributable to P21 in WT MEFs is modest and already near maximal, so that the 5-fold increase caused by IMP2 deletion has little further inhibitory effect. We have not however examined the effect of P21 overexpression in WT MEFs, because that would be an inappropriate model for IMP2^-/-^ MEFs, which exhibit numerous changes in cell cycle regulators other than P21, in directions that could either speed or slow proliferation (see Supplementary file 2, tab 3). Consequently, we have not pursued the question of why reducing P21 has a small effect of similar magnitude in both WT and IMP2 null MEFs, nor do we think that any of our explanations are sufficiently compelling to merit discussion.

The authors comment in two places in the manuscript that the changes in proliferation could be in part mediated through insulin receptor A. Since Foti et al. (Nature Medicine 2005) reported that knocking out HMG A1 markedly downregulated insulin receptor expression and that transfection of the cells with HMG A1 restored insulin receptor expression one would expect the manipulations herein to alter insulin receptor number. If these manipulations of HMGA1 do change insulin receptor expression then tumor proliferation could be driven through this receptor. Therefore, showing the change in insulin receptor number would strongly support their hypothesis that this is the mechanism mediating tumor cell proliferation under these conditions. It seems reasonable to measure insulin receptor expression under conditions of high and low HMGA1 expression.

As seen in Author response image 2 upper left, shRNA induced depletion of HMGA1 from WT MEFs reduces the abundance of InsR-B and InsR-A mRNA, determined by Q-PCR, ~four fold, in confirmation of several reports from the Brunetti and colleagues (Foti et.al., 2005; Liritano et.al., 2012); reciprocally, (lower left) doxycycline-induced overexpression of HMGA1 in both WT (lanes 1,3) and IMP2 null (lanes 2,4) MEFs increases InsR-B mRNA and InsR-A in MEFs ~2.1-2.6 fold. In contrast, deletion of IMP2 from MEFs (lower left, lanes 3,4), although reducing HMGA1 to a similar extent as accomplished by HMGA1-directed shRNA, has no effect on insR-A or InsR-B mRNA levels.

Similarly, in RD cells (Author response image 2, right), although doxycycline induction of HMGA1 increases InsR mRNA levels ~3-3.5 fold, the marked reduction in HMGA1 caused by CRISPR-mediated deletion of IMP2 does not alter the abundance of InsR A or B mRNA.

**Author response image 2. respfig2:** Depletion of HMGA1 reduces InsR mRNA whereas reduction or elimination of IMP2, although reducing HMGA1 abundance, does not alter InsR abundance.

We do not know the nature of the compensatory mechanisms that offset the reduction in InsR gene expression likely to have resulted from the reduced HMGA1 abundance caused by IMP2 deletion. Nevertheless, this data indicates clearly that the slowed proliferative rate of IMP2 depleted cells cannot be attributed to a reduced InsR abundance. Similarly, as shown in Figure 2, neither does IMP2 deletion alter the abundance of the IGF1R, but only the ability of IGF1R to be activated by IGF1, which in the experiment shown in Figure 2 we attribute predominantly to the upregulated expression of Grb14 and in cell culture proliferation to IGFBP2 as well.

7) The authors should discuss the data of Foti et al. and how they apply to their system. Additionally, those authors (Liritano et al. Mol endo 2012) have published data indicating that with HMG A1 downregulation there is enhanced insulin/IGF-I sensitivity which, if discussed, would also contribute to a more complete consideration of their hypothesis.

The metabolic phenotype of the HMGA1 KO mice as described by Foti et al., 2005 and pursued in the followup paper from this group (Liritano et.al., Mol Endo 2012; 26:1578) remains inexplicable. There is a reduction in insulin receptor abundance in the KO, but serum insulin levels are lower than in WT and glucose tolerance is minimally impaired; moreover, the hypoglycemic effect of injected insulin is actually greater in the KO. Liritano et.al, show that HMGA1 deletion reduces expression of the IGF binding proteins 1 and 3; IGFBP3 binds ~60-70% of circulating IGF1/2, IGFBP1 ~5% and most of the rest is bound by IGFBP2 (which our current paper shows is greatly increased by HMGA1 KO in cell culture). Liritano finds ~3X increase in unbound/free IGF-1in HMGA1 KO mice and reports modestly enhanced glycemic lowering to exogenous IGF-1 in the KO mice. Unfortunately, inasmuch as insulin does not bind the IGFBPs or IGF1R, these findings do not explain why there is a greater hypoglycemic effect of insulin in the HMGA1 KO despite reduced InsR expression. These complexities and unexplained phenomena discouraged us from dwelling on this data in our Discussion. Nevertheless, we have added a paragraph at the end of the Discussion that summarizes the information we consider defensible regarding the relationship of the components of the tumor promoting pathway we describe herein (i.e., HMGA2, IMP2, IGF2, HMGA1 and Grb14) to type 2 diabetes and the increased risk for cancers associated with that condition.

References:

1. Frasca F,1 Pandini G, Scalia, P, Sciacca, L, Mineo R, CostantinoA, Goldfine ID,

Belfiore A and Vigneri R. Insulin Receptor Isoform A, a Newly Recognized, High-

Affinity Insulin-Like Growth Factor II Receptor in Fetal and Cancer Cells. Mol Cell Biol.

1999. 19: 3278–3288.PMCID: PMC84122

2. Rowzee AM, Ludwig DL, Wood TL. Insulin-like growth factor type 1 receptor and

insulin receptor isoform expression and signaling in mammary epithelial cells.

Endocrinology. 2009.15:3611-9. doi: 10.1210/en.2008-1473.